# The cooperative action of CSB, CSA, and UVSSA target TFIIH to DNA damage-stalled RNA polymerase II

Yana van der Weegen [1], Hadar Golan-Berman[2], Tycho E. T. Mevissen [3], Katja Apelt[1], Román González-Prieto [4], Joachim Goedhart [5], Elisheva E. Heilbrun[2], Alfred C. O. Vertegaal[4], Diana van den Heuvel[1], Johannes C. Walter [3], Sheera Adar[2] & Martijn S. Luijsterburg [1✉]

The response to DNA damage-stalled RNA polymerase II (RNAPIIo) involves the assembly of the transcription-coupled repair (TCR) complex on actively transcribed strands. The function of the TCR proteins CSB, CSA and UVSSA and the manner in which the core DNA repair complex, including transcription factor IIH (TFIIH), is recruited are largely unknown. Here, we define the assembly mechanism of the TCR complex in human isogenic knockout cells. We show that TCR is initiated by RNAPIIo-bound CSB, which recruits CSA through a newly identified CSA-interaction motif (CIM). Once recruited, CSA facilitates the association of UVSSA with stalled RNAPIIo. Importantly, we find that UVSSA is the key factor that recruits the TFIIH complex in a manner that is stimulated by CSB and CSA. Together these findings identify a sequential and highly cooperative assembly mechanism of TCR proteins and reveal the mechanism for TFIIH recruitment to DNA damage-stalled RNAPIIo to initiate repair.

[1] Department of Human Genetics, Leiden University Medical Center, Einthovenweg 20, 2333 ZC Leiden, the Netherlands. [2] Department of Microbiology and Molecular Genetics, The Institute for Medical Research Israel–Canada, The Faculty of Medicine, The Hebrew University of Jerusalem, Ein Kerem, Jerusalem 91120, Israel. [3] Howard Hughes Medical Institute and Department of Biological Chemistry and Molecular Pharmacology, Harvard Medical School, Boston, MA 02115, USA. [4] Department of Cell and Chemical Biology, Leiden University Medical Center, Einthovenweg 20, 2333 ZC Leiden, the Netherlands. [5] Swammerdam Institute for Life Sciences, Section of Molecular Cytology, van Leeuwenhoek Centre for Advanced Microscopy, University of Amsterdam, Science Park 904, 1098 XH Amsterdam, the Netherlands. ✉email: m.luijsterburg@lumc.nl

Nucleotide excision repair (NER) is a versatile DNA repair pathway that removes a wide range of helix-distorting DNA lesions from the genome, including ultra-violet (UV) light-induced photolesions. Transcription-coupled repair (TCR) is a specialized NER sub-pathway that specifically removes DNA lesions from actively transcribed DNA strands[1]. It is believed that the TCR pathway is initiated by the stalling of elongating RNA polymerase II (RNAPIIo) at DNA lesions, which triggers the recruitment of the core NER machinery to repair these lesions[2]. After lesion recognition, the transcription factor IIH (TFIIH) complex is recruited to unwind the DNA[3,4], followed by dual incision, and the release of a 22–30 nucleotide-long DNA strand containing the lesion[5,6]. The generated single-stranded DNA gap is filled by repair synthesis and the nick is sealed[2]. However, the mechanism through which TCR recognizes transcription-blocking lesions and recruits the repair machinery is unknown.

Inherited defects that selectively impair TCR give rise to Cockayne Syndrome (CS) and UV-sensitive syndrome (UV$^S$S). Although cells from both CS and UV$^S$S patients show a defect in TCR[7,8], the phenotypes associated with these disorders are very different. CS is characterized by severe and progressive neurodegeneration[9,10], while UV$^S$S is characterized by mild UV sensitivity[11–13]. The majority of CS patients carry mutations in the *CSB* or *CSA* genes[14,15], while UV$^S$S patients carry mutations in the *UVSSA* gene[16,17].

The CSB protein contains a central SWI2/SNF2-like DNA-dependent ATPase domain[18], and resides in a complex with RNAPIIo[19,20]. Live-cell imaging suggests that CSB monitors the progression of transcription elongation by continuously probing RNAPIIo complexes[21]. It has been suggested that CSB is involved in the repositioning of RNAPII to make the DNA lesion accessible for repair proteins[22]. Although the association of CSB with RNAPII is sufficient to recruit TFIIH in vitro[23], it is unknown whether additional factors are required to trigger the recruitment of the repair machinery in vivo.

Like CSB, the CSA and UVSSA proteins also associate with DNA damage-stalled RNAPIIo[16,17,24,25]. The CSA protein contains seven WD40 repeats that form a seven bladed β-propeller[26]. Earlier work has shown that CSA is incorporated into a DDB1-CUL4-based E3 ubiquitin ligase complex[24,27] that becomes transiently activated in response to UV irradiation and targets CSB for proteasomal degradation[28]. Furthermore, the CSA complex also targets the UV-induced transcription repressor ATF3 as a means to regulate transcription restart after UV[29]. Current models suggest that CSA is dispensable for the recruitment of the excision repair machinery to stalled RNAPII[30], and that CSA is unlikely to recruit UVSSA to sites of UV-induced DNA damage[31]. Thus, the precise recruitment mechanism and the role of CSA in TCR is currently not clear.

The UVSSA protein contains an N-terminal VHS domain and a C-terminal DUF2043 domain of unknown function. Several studies reported that UVSSA, likely through its binding partner USP7, protects CSB from UV-induced degradation[16,17,25,32]. However, ectopic expression of CSB in UVSSA-deficient cells did not rescue TCR, suggesting that UVSSA has additional functions in this repair mechanism[16]. Moreover, UVSSA was found to associate with RNAPII[17,25], but whether UVSSA is constitutively bound to RNAPII, or associates with DNA damage-stalled RNAPII through either CSA or CSB is still a topic of debate.

The TFIIH complex consists of seven core subunits, including the XPB and XPD helicases, and three CAK kinase subunits[33]. While the CAK complex is crucial during transcription initiation, it inhibits the XPD helicase activity required for repair[34]. The release of the CAK complex from core TFIIH is triggered by the association of repair factors XPA and XPG, which switches TFIIH from a transcription factor into a repair factor[34,35].

Despite the knowledge that CSB, CSA, and UVSSA are required for TCR, we still know very little about how the interplay between these proteins targets the core repair machinery, including TFIIH, to DNA damage-stalled RNAPII. In this study, we demonstrate a sequential and highly cooperative assembly of TCR proteins and unveil the mechanism for TFIIH recruitment to DNA damage-stalled RNAPIIo.

## Results

**Isolation of active TCR complexes under native conditions.** Our current understanding of the assembly and functioning of multi-protein complexes that mediate transcription-coupled DNA repair (TCR) is fairly limited. This is largely due to a lack of sensitive methods to isolate active TCR complexes and analyze their composition. To overcome this limitation, we set out to establish a new immunoprecipitation-based method to isolate the elongating form of RNAPII (RNAPIIo) and associated proteins from the chromatin fraction of UV-irradiated cells under native conditions (Fig. 1a). To this end, we employed extensive benzonase treatment to solubilize the chromatin fraction after centrifugation, followed by immunoprecipitation using antibodies that recognize the Ser2-phosphorylated form of RNAPII. This RNAPII modification is absent from transcription start sites (TSS), but increases across gene bodies and is associated with transcription elongation[36]. Immunoprecipitation of RNAPIIo revealed a UV-specific association with the CSB and CSA proteins, as well as with CUL4, DDB1 and RBX1, which are subunits of a DDB1-CUL4 (CRL4) E3 ubiquitin ligase complex that associates with CSA[24,27]. In addition, several subunits of the TFIIH complex (XPD/p80, XPB/p89, GTF2H1/p62, and CDK7) also associated with RNAPIIo after UV (Fig. 1b; Supplementary Fig. 1a). Similar results were obtained when using antibodies against Ser5-phosphorylated RNAPII (Supplementary Fig. 1b). Importantly, we did not detect an RNAPII–TFIIH interaction in unirradiated cells, suggesting that our procedure indeed does not capture RNAPII involved in transcription initiation during which it interacts extensively with TFIIH[37]. In line with this, genomic DNA fragments immunoprecipitated together with RNAPIIo after UV irradiation were highly enriched for the most abundant UV-induced DNA lesion (cyclobutane pyrimidine dimer; CPD) (Fig. 1c). These findings suggest that a considerable fraction of the RNAPIIo molecules we capture under our conditions are stalled at DNA lesions. Although the CS proteins and TFIIH readily assembled with RNAPIIo after UV irradiation, downstream repair proteins such as XPA, ERCC1-XPF, and XRCC1 could not be detected (Fig. 1b, Supplementary Fig. 1a). It should be noted that we could not detect UVSSA either after pull-down of RNAPIIo or in whole cell lysates due to a lack of specific antibodies (Supplementary Fig. 1c). These initial results suggest that CSB, CSA, and TFIIH associate with DNA damage-stalled RNAPII, but that the assembly of downstream repair factors may require the removal or backtracking of RNAPII to make the lesion accessible to the repair machinery[22].

**CSA is recruited to DNA damage-stalled RNAPII by CSB.** To acquire more insights into the initial assembly of TCR factors, we generated CSB, CSA, and UVSSA knockout (KO) cells using CRISPR-Cas9-mediated genome editing in U2OS cells equipped with the Flp-In/T-REx system. The knockout of CSB, CSA, and UVSSA was confirmed by western blot analysis and/or DNA sequencing (Fig. 1d; Supplementary Fig. 2a, b). Clonogenic survival assays revealed that all TCR-KO cells were highly sensitive to transcription-blocking DNA damage induced by Illudin S (Fig. 1e), which is a natural compound from mushroom *O. illudens* causing DNA lesions that are exclusively repaired by TCR[38].

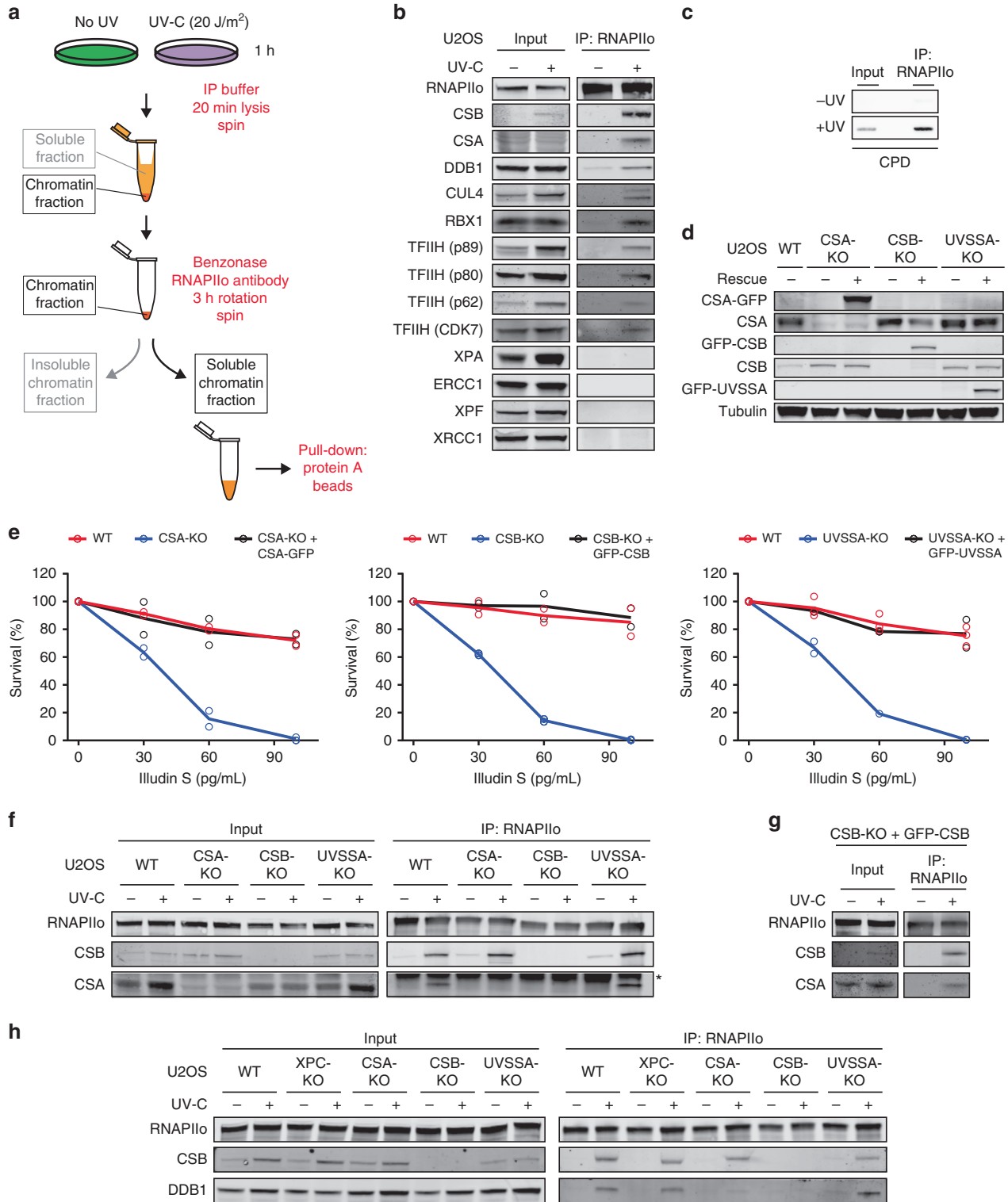

**Fig. 1 CSA is recruited to DNA damage-stalled RNAPIIo by CSB. a** Outline of a new IP method to isolate RNAPIIo and associated proteins from mock-treated or UV-irradiated (20 J/m²) U2OS (FRT) cells. **b** Endogenous RNAPII Co-IPs on WT cells stained for the indicated TCR proteins. Note that it is not possible to stain for all these proteins on one membrane. This panel is a composite of several representative Co-IPs. See Supplementary Fig. 1a for each individual Co-IP. **c** Endogenous RNAPII Co-IP followed by slot blot analysis of CPDs **d** Western blot analysis of CSA, CSB, and UVSSA knockout cells complemented with inducible GFP-tagged versions of these proteins (*n* = 2). See Supplementary Fig. 2a, b for validation of knockouts by sequencing. **e** Clonogenic Illudin S survival of WT, CSA, CSB, and UVSSA knockout and rescue cell lines. Each symbol represents the mean of an independent experiment (*n* = 2 for all except for WT in UVSSA-KO figure which is *n* = 3) each experiment contains two or three technical replicates. Endogenous RNAPII Co-IP on **f** WT, CSA, CSB, and UVSSA knockout cells, **g** CSB-KO stably expressing GFP-CSB, and **h** WT, XPC, CSA, CSB, and UVSSA knockout cells. The asterisk in **e** indicates the heavy chain of the RNAPII antibody. At least two independent replicates of each IP experiment were performed obtaining similar results. Source data are provided as a Source Data file.

Importantly, complementation of these TCR-KO cells with inducible GFP-tagged versions of CSB, CSA, and UVSSA fully restored their resistance to Illudin S (Fig. 1d, e). We next applied our immunoprecipitation-based method in the different TCR-KO cells to establish how CSB and CSA recruitment to DNA damage-stalled RNAPIIo is regulated. CSB associated with RNAPIIo in wild-type (WT), CSA-KO, and UVSSA-KO cells specifically after UV irradiation, suggesting that CSB is the first of these proteins to associate with DNA damage-stalled RNAPIIo (Fig. 1f). The association of CSA with stalled RNAPIIo was abolished in CSB-KO cells, but was not affected in cells lacking UVSSA (Fig. 1f). Importantly, re-expressing GFP-tagged CSB in the CSB-KO cells restored the association between RNAPIIo and CSA (Fig. 1f, g). Next, we asked whether CSA mediates the recruitment of the CRL4 complex, including its E3 ubiquitin ligase partner DDB1 to DNA damage-stalled RNAPIIo. As an additional control we also included XPC-KO cells, which are deficient in global genome repair (GGR; Supplementary Fig. 2c). Immunoprecipitation of RNAPIIo revealed a UV-specific interaction with DDB1 in WT, XPC-KO, and UVSSA-KO cells (Fig. 1h). However, this interaction was completely abolished in CSA-KO and CSB-KO cells, showing that CSA indeed mediates the recruitment of the CRL4 complex to lesion-stalled RNAPIIo (Fig. 1h).

**Mapping the CSA-interaction motif (CIM) in CSB.** In order to gain a better understanding of the CSA recruitment mechanism by CSB, we aimed to identify the region in CSB that is required for the interaction with CSA. To this end, we employed a chromatin-tethering approach making use of the U2OS 2-6-3 cell line harboring an integrated LacO array in the genome[39]. This cell line enables the analysis of protein–protein interactions by tethering proteins of interest fused to the bacterial LacR and fluorescent protein mCherry to a defined chromosomal region[40,41] (Fig. 2a). Expression of mCherry-LacR fused to full-length CSB (Fig. 2b) resulted in clear localization of the fusion protein to the LacO array and triggered the robust recruitment of CSA-GFP (Fig. 2c). In contrast, expression of LacR alone failed to recruit CSA-GFP to the LacO array (Fig. 2c).

To identify the CSA-interaction domain in CSB, we fused various truncated fragments of CSB to mCherry-LacR and examined their ability to recruit CSA-GFP to the LacO array (Fig. 2b, Supplementary Figs. 3, 4). Fragments of CSB spanning the N-terminus or the central region containing the conserved ATPase/helicase domain (N, M, and ΔC) were unable to recruit CSA-GFP. Conversely, tethering of a LacR-tagged CSB region spanning the C-terminus (C and ΔN) triggered robust recruitment of CSA-GFP (Fig. 2b–d, Supplementary Fig. 3). These results suggest that the C-terminus of CSB is essential for the interaction with CSA. The C-terminus of CSB contains a ubiquitin-binding domain (UBD; 1400–1428[42]) and a recently identified winged-helix domain (WHD; 1417–1493) that interacts with RIF1[43]. Interestingly, we found that the most N-terminal region (1221–1305) of the CSB C-terminus alone, or fragments containing solely the UBD (1400–1493) or WHD (1417–1493) domains do not support CSA recruitment. However, a region just upstream of the UBD (1306–1399) is sufficient to mediate CSA recruitment to the LacO array (Fig. 2b–d, Supplementary Fig. 3). Importantly, we found that tethering full-length CSB lacking this minimal interaction region (Δ1306–1399) indeed failed to support CSA recruitment (Fig. 2b–d). Further deletion analysis showed that CSB lacking the region just upstream of the UBD (1353–1399) failed to recruit CSA-GFP, whereas CSB lacking the UBD (1400–1428) or amino acids 1306–1352 were fully proficient in interacting with CSA-GFP (Supplementary Fig. 4). Moreover, while CSB$^{\Delta 1353-1368}$ and CSB$^{\Delta 1369-1384}$ were fully proficient in

recruiting CSA-GFP to the LacO array, deleting amino acids 1385–1399 abolished the ability of CSB to interact with CSA-GFP (Fig. 2b–d, Supplementary Fig. 4). These findings identify an evolutionary conserved CSA-interaction motif (CIM) in CSB that is located between amino acids 1385–1399 (Fig. 2e; Supplementary Fig. 5).

**The C-terminal CIM in CSB recruits CSA to DNA damage-stalled RNAPII.** We next set out to address the importance of this new CSB motif under more physiological conditions. To this end, we stably expressed GFP-tagged CSB$^{WT}$ or CSB$^{\Delta CIM}$ in CSB-KO cells (Fig. 3a, b). Pull-down of GFP-tagged CSB$^{WT}$ showed a strong UV-induced interaction with CSA, which was virtually absent after pull-down of CSB$^{\Delta CIM}$ even though equal amounts of CSB were immunoprecipitated (Fig. 3c). These findings were confirmed by quantitative mass spectrometry (MS) after pull-down of GFP-tagged versions of either CSB$^{WT}$ or CSB$^{\Delta CIM}$ (Supplementary Fig. 6a–c, see Supplementary Table 7 for link to interactive volcano plots). Immunoprecipitation of endogenous RNAPIIo in these cell lines showed that both CSB$^{WT}$ and CSB$^{\Delta CIM}$ associated equally with RNAPIIo after UV irradiation. However, CSB$^{\Delta CIM}$ failed to recruit CSA to DNA damage-stalled RNAPIIo, while a strong association of CSA was observed in cells expressing CSB$^{WT}$ (Fig. 3d). Importantly, the stable expression of GFP-CSB$^{\Delta CIM}$ in CSB-KO cells failed to restore sensitivity to Illudin S, while expression of GFP-CSB$^{\Delta WT}$ almost fully rescued this phenotype (Fig. 3e). To determine whether the CIM can mediate a functional interaction between CSB and CSA, we mixed recombinant *Xenopus laevis* CSB$^{WT}$ or CSB$^{\Delta CIM}$ with ubiquitin, E1, E2, and the E3 ubiquitin ligase CRL4$^{CSA}$ consisting of *Xenopus laevis* CSA, DDB1, CUL4A, and RBX1 (Supplementary Fig. 6d). While xlCRL4$^{CSA}$ promoted the efficient ubiquitylation of xlCSB$^{WT}$, it did not ubiquitylate xlCSB$^{\Delta CIM}$ (Fig. 3f). These data suggest that xlCSB uses its CIM to interact directly with xlCSA. Consistent with this interpretation, immobilized xlCSB$^{WT}$ but not xlCSB$^{\Delta CIM}$ interacted with endogenous xlCSA from *Xenopus* egg extract (Fig. 3g). Similar results were observed when xlCSB was substituted with hsCSB (Fig. 3f, g). Collectively, these data demonstrate that CSA is recruited to DNA damage-stalled RNAPIIo by CSB through direct interactions with the newly identified C-terminal CIM in CSB.

**UVSSA is recruited to DNA damage-stalled RNAPIIo by CSA.** Previous studies have demonstrated that UVSSA associates with RNAPIIo, but due to conflicting results, it remains unclear if UVSSA recruitment to RNAPIIo is enhanced by UV irradiation and dependent on the CS proteins[17,25,31]. Therefore, we monitored GFP-UVSSA recruitment to RNAPIIo in UVSSA-KO cells complemented with GFP-UVSSA (WT) in which we additionally knocked out either CSB or CSA. The knockout of CSB and CSA was verified by western blot analysis, DNA sequencing (Fig. 4a; Supplementary Fig. 2), and Illudin S clonogenic survival assays (Fig. 4b). Immunoprecipitation of endogenous RNAPIIo in these cell lines showed that GFP-UVSSA became readily detectable after UV irradiation in WT cells, whereas this interaction was virtually absent in CSA-KO and CSB-KO cells (Fig. 4c). Thus, GFP-UVSSA is targeted to DNA damage-stalled RNAPIIo in a manner that is dependent on the CS proteins[17]. Moreover, pull-down of GFP-UVSSA confirmed a robust UV-induced association with RNAPIIo, CSB, and CSA. However, these UV-specific interactions were abolished in CSB-KO and CSA-KO cells. Interestingly, we detected a weak UV-independent interaction between GFP-UVSSA and CSA, which was enhanced after UV irradiation in a manner that required CSB (Fig. 4d). These

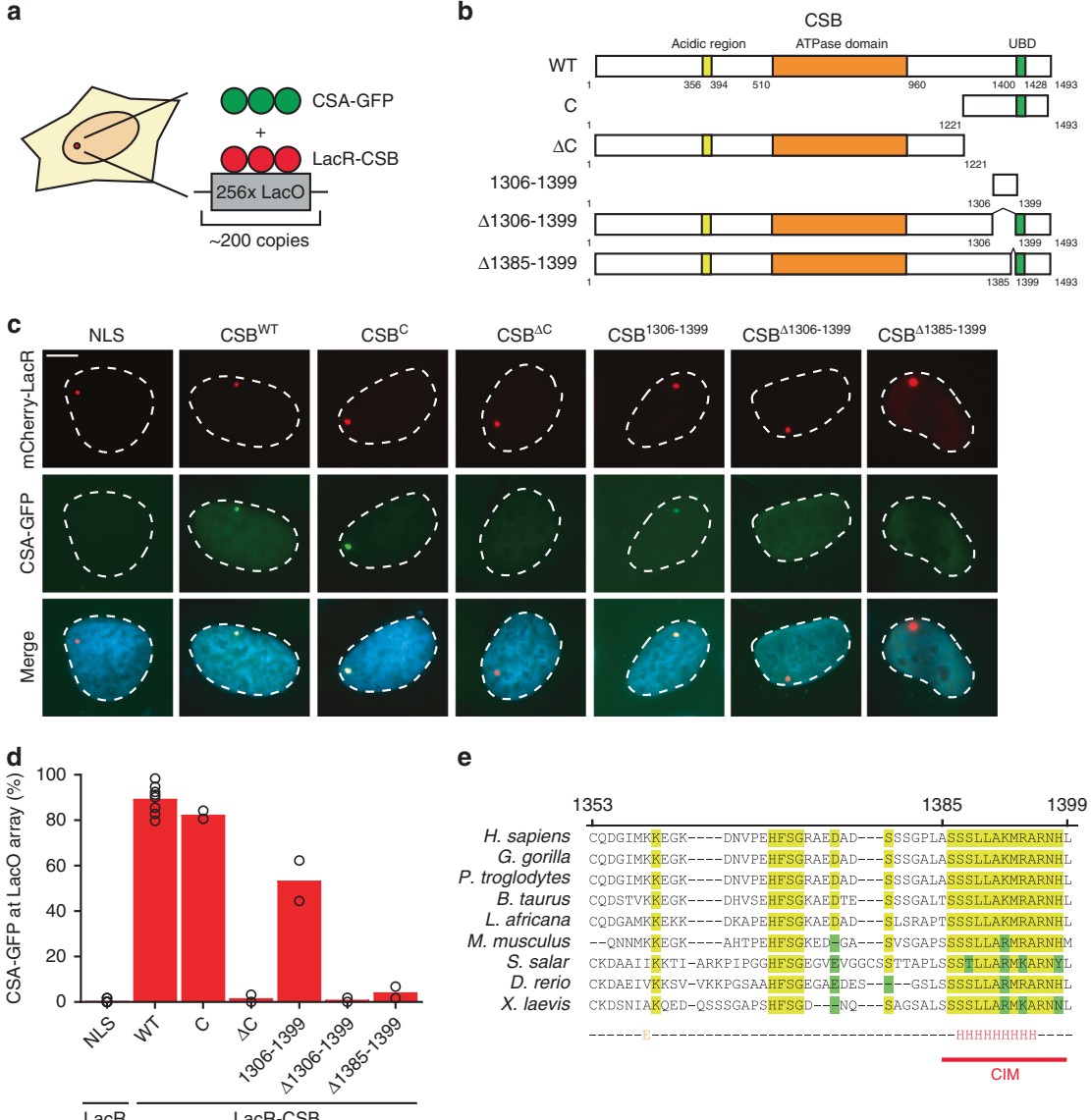

**Fig. 2 CSA interacts with the newly identified C-terminal CIM of CSB. a** Outline of the chromatin-tethering approach in U2OS 2-6-3 cells. **b** A schematic representation of CSB and its deletion mutants. **c** Recruitment of CSA-GFP to the LacO array upon tethering of the indicated mCherry-LacR fusion proteins (scale bar = 5 μm). See Supplementary Figs. 3 and 4 for a full overview of all tested mutants. **d** Quantification of CSA-GFP and mCherry-LacR-CSB co-localization at the LacO array. Each symbol represents the mean of an independent experiment ($n = 2$ for all except LacR-NLS and LacR-CSB$^{WT}$ which is $n = 8$, >50 cells collected per experiment). **e** Sequence alignment of CSB orthologues. See Supplementary Fig. 5 for additional alignments. Source data are provided as a Source Data file.

findings suggest that the cooperative assembly of the TCR complex is important to mediate efficient targeting of UVSSA to lesion-stalled RNAPIIo.

**CSB and CSA are required for the recruitment of the TFIIH complex.** CSB, CSA, and UVSSA can each associate with TFIIH[23,26,44], but which of these proteins is responsible for the recruitment of TFIIH to DNA damage-stalled RNAPIIo is currently unknown. To directly asses if CSB and CSA are required for the recruitment of TFIIH, we monitored TFIIH (p62 and p89) recruitment in UVSSA-KO complemented with GFP-UVSSA (WT) in which we additionally knocked out either CSB or CSA. Immunoprecipitation of endogenous RNAPIIo revealed a UV-specific interaction with TFIIH in WT cells, while these interactions were severely reduced in the CSB-KO and CSA-KO cells (Fig. 5a). Interestingly, TFIIH also failed to associate with

RNAPIIo in CSB-KO cells complemented with GFP-CSB$^{ΔCIM}$ (Supplementary Fig. 6e), consistent with our findings that this mutant is not capable of recruiting CSA (Fig. 3c, g). These initial results suggest that the TFIIH complex is recruited in a manner that requires both CS proteins.

**UVSSA targets the TFIIH complex to DNA damage-stalled RNAPIIo.** It has been reported that UVSSA can interact with TFIIH[16,32,44], but whether this reflects a constitutive interaction or a UV-induced association is unclear. To gain more insight into the nature of this interaction, we immuno-precipitated GFP-UVSSA from the solubilized chromatin fraction of mock-treated and UV-irradiated cells followed by mass spectrometry (MS). Following UV irradiation, our MS analysis identified 28 UV-specific UVSSA interactors, including CSB, the CSA-interacting protein DDB1, and RNAPII subunits.

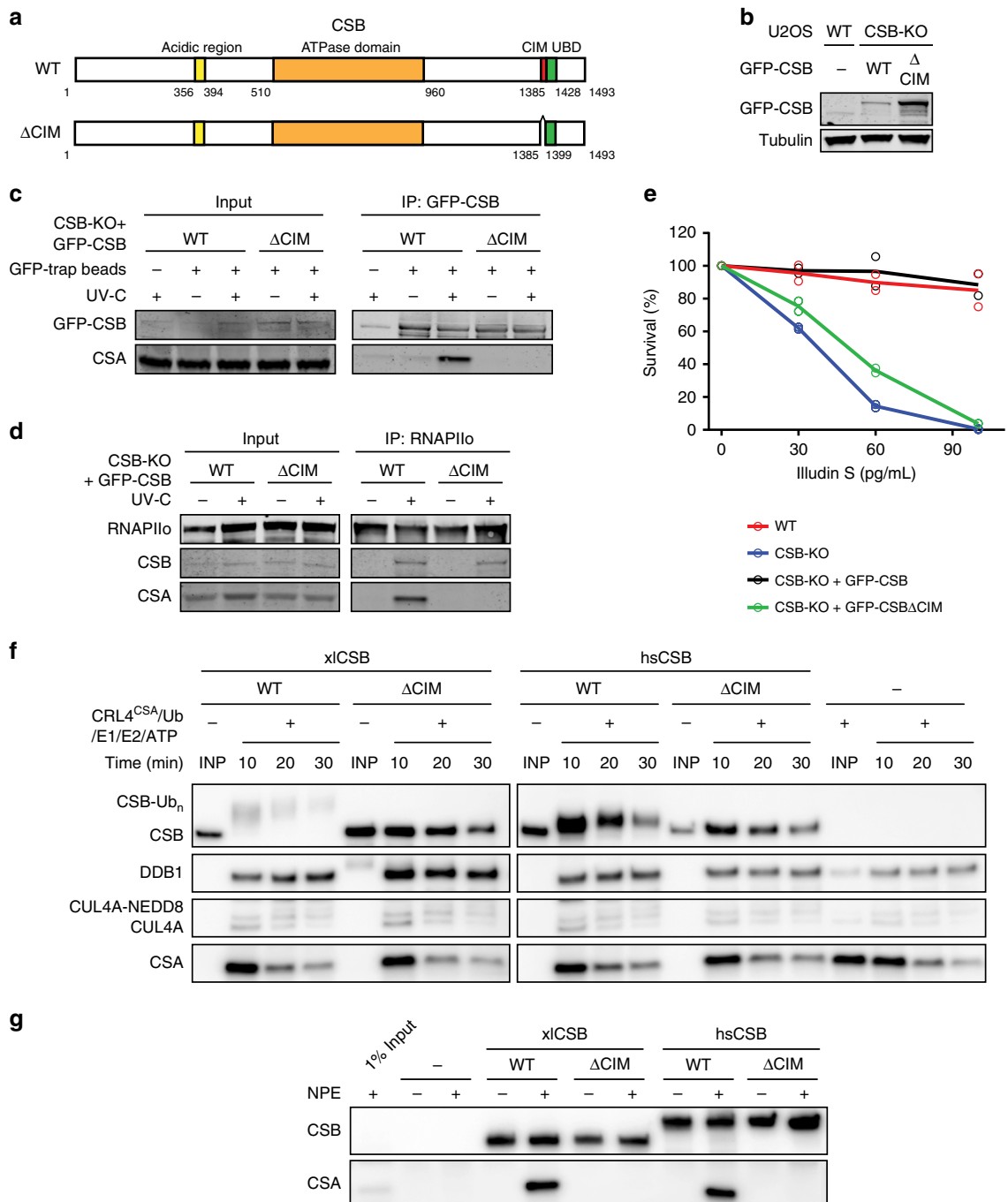

**Fig. 3 The CIM of CSB mediates the recruitment of CSA to DNA damage-stalled RNAPIIo. a** A schematic representation of CSB and the CSB$^{\Delta CIM}$ mutant. **b** Western blot analysis of U2OS (FRT) and CSB-KO complemented with either GFP-CSB$^{WT}$ or GFP-CSB$^{\Delta CIM}$ ($n = 2$). **c** Co-IP of GFP-CSB$^{WT}$ and GFP-CSB$^{\Delta CIM}$ on the combined soluble and chromatin fraction. **d** Endogenous RNAPII Co-IP in GFP-CSB$^{WT}$ and GFP-CSB$^{\Delta CIM}$ cell lines. See also Supplementary Fig. 6e for additional Co-IP data. **e** Clonogenic Illudin S survival of WT and CSB-KO cell lines and the GFP-tagged CSB rescue cell lines. Each symbol represents the mean of an independent experiment ($n = 2$), each of which is based on two technical replicates. Note that the same survival data for WT, CSB-KO and CSB-KO + GFP-CSB is also shown in Fig. 1e. **f** In vitro ubiquitylation of recombinant *Xenopus laevis* (xl) and *Homo sapiens* (hs) CSB variants with recombinant xlCRL4$^{CSA}$, E1, E2, ubiquitin, and ATP. At indicated times, in vitro ubiquitylation reactions were stopped and blotted with anti-FLAG (top three panels) or anti-xlCSA (bottom panel) antibodies. See also Supplementary Fig. 6d. **g** Immobilized recombinant CSB variants were incubated with *Xenopus laevis* nucleoplasmic extract (NPE), recovered, and blotted with anti-FLAG (top panel) or anti-xlCSA (bottom panel) antibody. At least two independent replicates of each IP and in vitro ubiquitylation experiment were performed obtaining similar results. Source data are provided as a Source Data file.

Additionally, among the most prominent UV-specific interactions were the TFIIH subunits XPB/p89 and XPD/p80 (Fig. 5b; see Supplementary Table 7 for link to interactive volcano plots). These findings demonstrate that UVSSA interacts in a UV-specific manner with TFIIH.

Immunoprecipitation of GFP-UVSSA indeed confirmed a UV-specific interaction with TFIIH subunits by western blot analysis (Fig. 5c). Strikingly, these interactions were severely reduced in the CSB-KO and CSA-KO cells, suggesting a cooperative interaction mechanism in which CSB is required to stabilize the

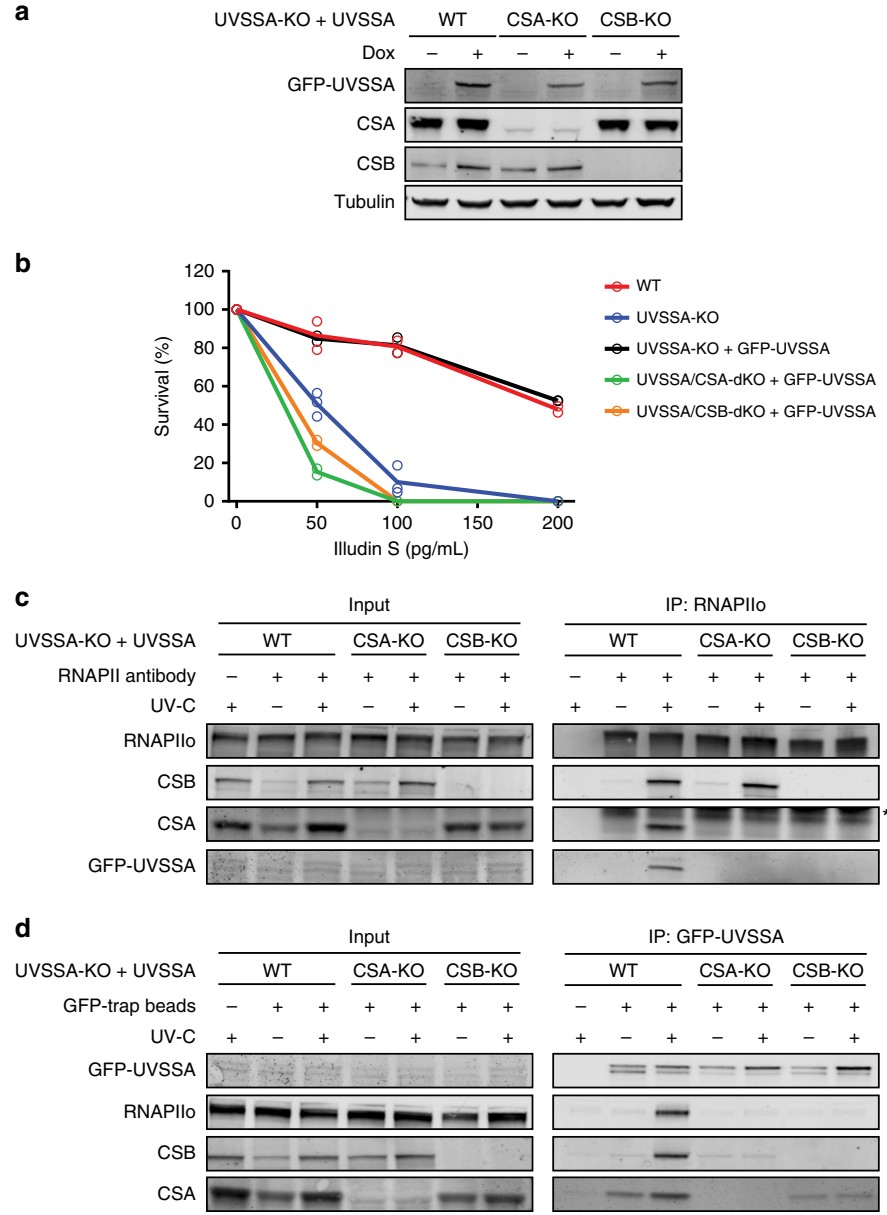

**Fig. 4 UVSSA is recruited to DNA damage-stalled RNAPIIo by CSA. a** Western blot analysis of UVSSA-KO, UVSSA/CSA-dKO, and UVSSA/CSB-dKO complemented with GFP-UVSSA ($n = 2$). **b** Clonogenic Illudin S survival of WT, UVSSA-KO, UVSSA/CSA-dKO, and UVSSA/CSB-dKO cell lines complemented with GFP-UVSSA. Each symbol represents the mean of an independent experiment ($n = 2$ for all except for UVSSA-KO which is $n = 3$), each of which is based on two technical replicates. **c** Endogenous RNAPII Co-IP on UVSSA-KO, UVSSA/CSA-dKO, and UVSSA/CSB-dKO complemented with GFP-UVSSA. **d** Co-IP of GFP-UVSSA in UVSSA-KO, UVSSA/CSA-dKO, and UVSSA/CSB-dKO cell lines. The asterisk in **c** indicates the heavy chain of the RNAPII antibody. At least two independent replicates of each IP experiment were performed obtaining similar results. Source data are provided as a Source Data file.

interaction between CSA and UVSSA, while CSA is required to stabilize the interaction between UVSSA and TFIIH.

We subsequently asked if UVSSA is also required for TFIIH recruitment. To this end, we employed our immunoprecipitation-based method in CSB-KO, CSA-KO, and UVSSA-KO cells to monitor TFIIH recruitment. In addition, we included XPA-KO cells (Supplementary Fig. 2c) as a positive control since XPA recruitment, at least during GGR, occurs downstream of TFIIH[45]. Immunoprecipitation of endogenous RNAPIIo in these cell lines revealed a UV-specific interaction with TFIIH in WT and XPA-KO cells (Fig. 5d). These findings suggest that XPA recruitment does not only occur downstream of TFIIH in GGR but also in TCR. Interestingly, similar to CSB-KO and CSA-KO

cells, we found that the UV-induced interaction between RNAPIIo and TFIIH was severely reduced in UVSSA-KO cells (Fig. 5d). Furthermore, complementation of these TCR-KO cells with inducible GFP-tagged versions of CSB, CSA, and UVSSA fully restored the UV-induced association of TFIIH to RNAPIIo (Fig. 5e). These findings demonstrate that CSB, CSA, and UVSSA are equally important for the recruitment of the TFIIH complex to DNA damage-stalled RNAPIIo.

**Genome-wide XR-seq confirms that UVSSA is a core TCR factor.** Our findings show that UVSSA, just like CSA and CSB, is required to recruit TFIIH to initiate TCR-mediated repair. To

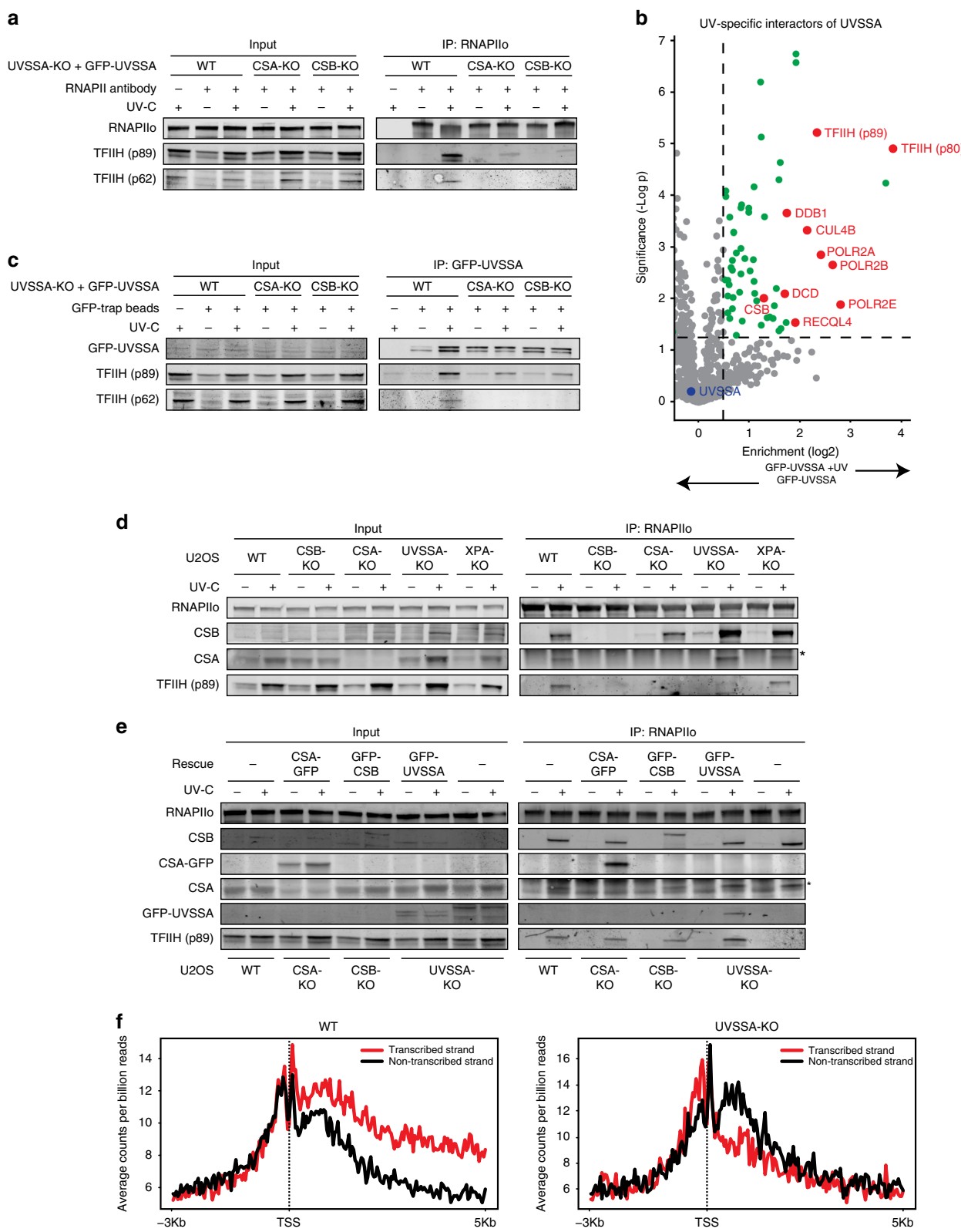

provide further support for a role of UVSSA in TCR, we carried out genome-wide XR-sequencing (XR-seq), which enables the generation of genome-wide repair maps by isolating and sequencing the 30-mers that are generated upon dual incision[46,47]. We generated nucleotide-resolution maps of UV-induced CPDs repair in U2OS WT cells (Fig. 5f; Supplementary

Fig. 7a; Supplementary Table 8), which revealed that CPD repair under these conditions is enriched on the transcribed strands within gene bodies consistent with TCR-mediated repair[46]. Importantly, the bias in CPD repair observed in transcribed strands was completely lost in both CSA-KO (Supplementary Fig. 7a) and UVSSA-KO cells (Fig. 5f). These findings provide

**Fig. 5 CSA, CSB, and UVSSA are equally important for TFIIH recruitment. a** Endogenous RNAPII Co-IP in UVSSA-KO, UVSSA/CSA-dKO, and UVSSA/CSB-dKO complemented with GFP-UVSSA. **b** Volcano plot depicting the statistical differences of the MS analysis on GFP-UVSSA pull-down in mock-treated and UV-irradiated samples. The enrichment (log$^2$) is plotted on the x-axis and the significance (t-test -log p value) is plotted on the y-axis. All significantly UV-induced hits are indicated in green. Several selected hits are shown in red (Link to the interactive volcano plots: GFP-UVSSA vs GFP-UVSSA + UV). **c** Co-IP of GFP-UVSSA in UVSSA-KO and UVSSA-dKO cells complemented with GFP-UVSSA. **d** Endogenous RNAPII Co-IP in WT, CSB-KO, CSA-KO, UVSSA-KO, and XPA-KO cells. **e** Endogenous RNAPII Co-IP in WT and UVSSA-KO cells and CSA-KO, CSB-KO, and UVSSA-KO cells complemented with GFP-tagged versions of these proteins. The asterisk in **d** and **e** indicates the heavy chain of the RNAPII antibody. **f** CPD XR-seq repair signal 3 Kb upstream and 5 Kb downstream of the annotated TSS of 16.088 genes in WT and UVSSA-KO cells. Signal is plotted separately for the transcribed (red) and non-transcribed (black) strands. The data represent the average of two independent experiments with a bin size of 40 nt. See also Supplementary Fig. 7a for additional XR-seq data. At least two independent replicates of each IP experiment were performed obtaining similar results. Source data are provided as a Source Data file.

direct genome-wide support for an essential role of UVSSA in TCR.

**UVSSA is the key protein that recruits TFIIH to DNA damage-stalled RNAPIIo.** We next asked whether TFIIH is recruited via direct protein–protein contacts with UVSSA, or whether CSB and CSA also contribute to this interaction. To address this, we generated UVSSA separation-of-function mutants that are selectively impaired in their interaction with either CSA (UVSSA$^{\Delta100-200}$) or the TFIIH complex (UVSSA$^{\Delta400-500}$)[32] (Fig. 6a). These separation-of-function mutants were characterized using our chromatin-tethering approach. mCherry-LacR-UVSSA$^{WT}$ clearly localized to the LacO array and triggered the robust recruitment of CSA-GFP and endogenous TFIIH (Fig. 6b, c). As expected, mCherry-LacR-UVSSA$^{\Delta100-200}$ was unable to recruit CSA-GFP to the LacO array, but triggered robust TFIIH recruitment (Fig. 6b, c). In contrast, mCherry-LacR-UVSSA$^{\Delta400-500}$ was unable to recruit TFIIH to the LacO array, but was proficient in recruiting CSA-GFP (Fig. 6b, c). These results confirm that UVSSA contains a CSA-interacting region (CIR; amino acids 100–200) and a TFIIH-interacting region (TIR; amino acids 400–500).

To elucidate the importance of the CIR and TIR in UVSSA under more physiological conditions, we stably expressed inducible GFP-UVSSA$^{WT}$, GFP-UVSSA$^{\Delta CIR}$, or GFP-UVSSA$^{\Delta TIR}$ in UVSSA-KO cells (Fig. 6d). Pull-down of GFP-UVSSA$^{WT}$ showed a strong UV-induced interaction with RNAPIIo, CSB, CSA, and TFIIH. These interactors were virtually absent after pull-down of GFP-UVSSA$^{\Delta CIR}$ (Fig. 6e; Supplementary Fig. 7b, c). The UVSSA$^{\Delta CIR}$ mutant was unable to interact with CSA and we found that its association with TFIIH was also abolished. This result is consistent with the finding that the UVSSA–TFIIH interaction is reduced in CSA-KO cells (Fig. 5c) and suggests that CSA stabilizes the interaction between UVSSA and TFIIH. Pull-down of GFP-UVSSA$^{\Delta TIR}$ resulted in a strong UV-induced interaction with RNAPIIo, CSB, and CSA, while its interaction with TFIIH was completely abolished (Fig. 6e; Supplementary Fig. 7b, c). These findings were largely confirmed by quantitative MS after pull-down of GFP-tagged versions of UVSSA. Interestingly, this approach also identified several proteins that interacted with UVSSA$^{WT}$, but not with the UVSSA mutants (Supplementary Fig. 8a–e; see Supplementary Table 7 for link to interactive volcano plots).

We next set out to directly asses the ability of these UVSSA mutants to participate in TCR complex assembly. Immunoprecipitation of endogenous RNAPIIo showed a UV-specific association of RNAPIIo with CSB and CSA in both UVSSA$^{WT}$ and mutant cell lines (Fig. 6f). Endogenous RNAPIIo immunoprecipitation revealed a UV-specific interaction with GFP-UVSSA$^{WT}$ and GFP-UVSSA$^{\Delta TIR}$, whereas GFP-UVSSA$^{\Delta CIR}$ failed to associate with RNAPIIo. The fact that a mutant of UVSSA that is deficient in its association with CSA fails to be recruited confirms our earlier findings that CSA is essential to recruit UVSSA to DNA damage-stalled RNAPIIo (Fig. 4c). Furthermore, the recruitment of TFIIH (p89) to DNA damage-stalled RNAPIIo was abolished in both UVSSA mutant cell lines (Fig. 6f). These experiments strongly suggest that TFIIH is recruited to DNA damage-stalled RNAPIIo via direct protein–protein contacts with UVSSA. Importantly, the stable expression of GFP-UVSSA$^{\Delta CIR}$ and GFP-UVSSA$^{\Delta TIR}$ in UVSSA-KO cells failed to restore their sensitivity to Illudin S, which was almost fully restored by GFP-UVSSA$^{WT}$ (Fig. 7a). In line with this, genome-wide XR-seq revealed that the stable expression of GFP-UVSSA$^{\Delta CIR}$ and GFP-UVSSA$^{\Delta TIR}$ in UVSSA-KO cells failed to restore the repair of CPDs in the transcribed strand, which was fully restored by GFP-UVSSA$^{WT}$ (Fig. 7b; Supplementary Fig. 9a; Supplementary Table 8). These findings confirm that both the CIR and the TIR of UVSSA have an essential role in TCR.

Altogether, our data reveals a sequential and cooperative assembly mechanism of the human TCR complex, which involves the stepwise assembly of CSB, CSA, and UVSSA to target the TFIIH complex to DNA damage-stalled RNAPIIo to initiate DNA repair (Fig. 7c).

## Discussion

Although it has been recognized for some time that CSA, CSB, and UVSSA are required for TCR, remarkably little is known about how these proteins cooperate to trigger TCR. Our findings suggest a highly cooperative recruitment mechanism that involves the sequential association of CSB, CSA, and UVSSA to target the TFIIH complex to DNA damage-stalled RNAPIIo to initiate repair.

We show that both CSB and CSA associate with RNAPIIo in a manner that is strongly induced by UV irradiation, and that CSA recruitment is completely dependent on CSB. This is in line with earlier work showing that CSB facilitates the translocation of CSA to the nuclear matrix after UV irradiation[48]. Moreover, we demonstrate that CSA is required for the association of DDB1 with RNAPIIo, suggesting that CSA is recruited to DNA damage-stalled RNAPIIo as part of a CRL4$^{CSA}$ complex[24,27]. Previous findings suggested that CSB dynamically associates with RNAPIIo under undamaged conditions and that this interaction is stabilized upon UV irradiation[21,49]. While our method may not be sensitive enough to capture these transient interactions, our findings do support that the CSB-RNAPIIo interaction is stabilized after UV irradiation.

Earlier observations suggested that CSB physically interacts with CSA[26,28], while other studies failed to detect this association[19,20]. Our findings fully support a direct UV-induced association between the CS proteins. Importantly, we identified the CIM in the C-terminus of CSB that is essential for targeting CSA to stalled RNAPIIo. Interestingly, the CIM region in CSB is evolutionary conserved in species that also contain the CSA gene, including mammals, amphibians and fish (Supplementary Fig. 5).

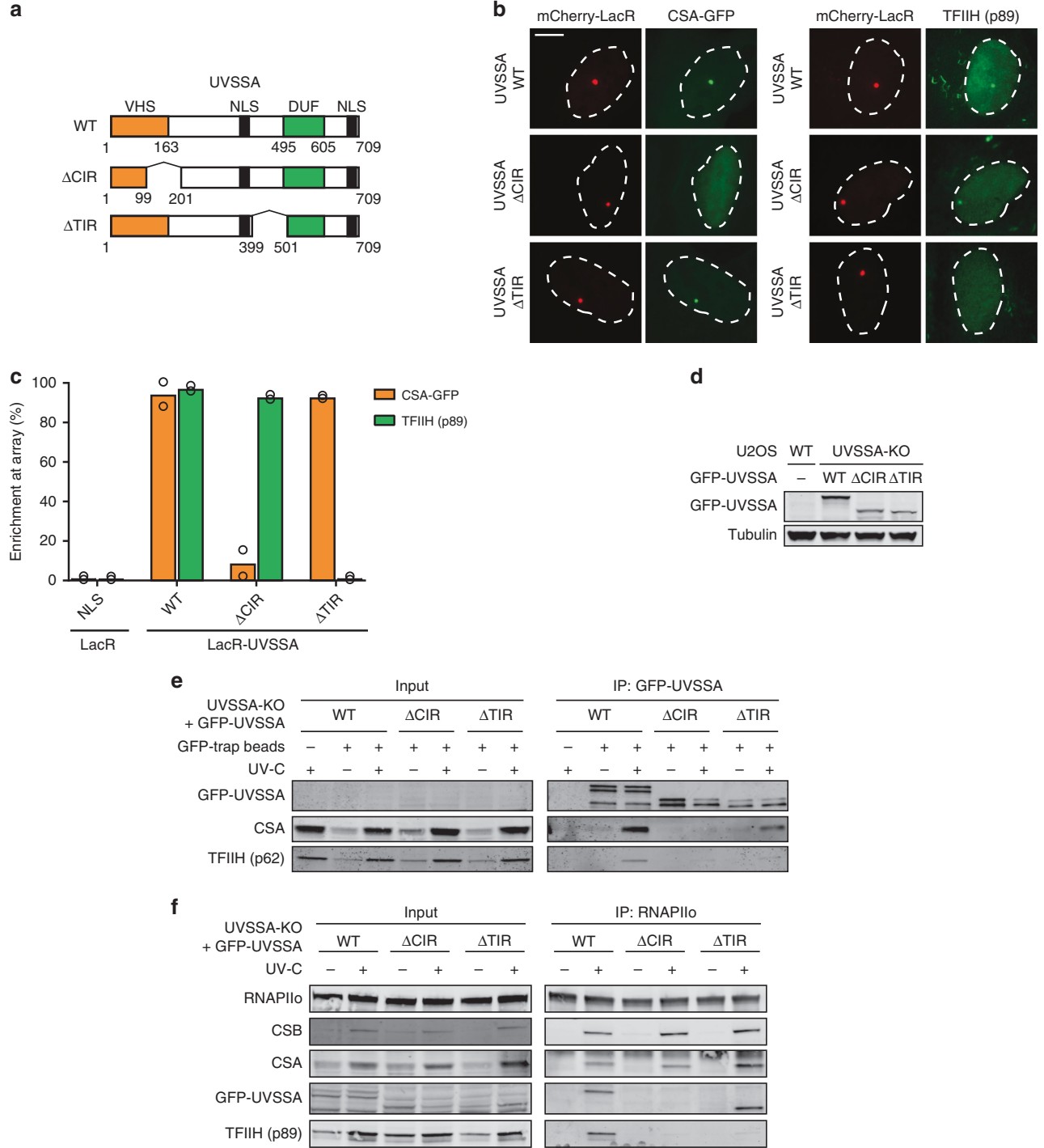

**Fig. 6 UVSSA is the key protein that recruits TFIIH. a** A schematic representation of UVSSA WT and deletion mutants. The CSA-interacting region (CIR) and TFIIH-interacting region (TIR) are indicated. **b** Recruitment of CSA-GFP and TFIIH (p89) to the LacO array upon tethering of the indicated mCherry-LacR fusion proteins (scale bar = 5 μm). **c** Quantification of CSA-GFP and endogenous TFIIH (p89) co-localization at the LacO array. Each symbol represents the mean of an independent experiment (*n* = 2, >50 cells collected per experiment). **d** Western blot analysis of U2OS (FRT) and UVSSA-KO cells complemented with GFP-UVSSA$^{WT}$, GFP-UVSSA$^{ΔCIR}$, and GFP-UVSSA$^{ΔTIR}$ (*n* = 2). **e** Co-IP of GFP-UVSSA$^{WT}$, GFP-UVSSA$^{ΔCIR}$, and GFP-UVSSA$^{ΔTIR}$. **f** Endogenous RNAPIIo Co-IP in GFP-UVSSA$^{WT}$, GFP-UVSSA$^{ΔCIR}$, and GFP-UVSSA$^{ΔTIR}$ cell lines. At least two independent replicates of each IP experiment were performed obtaining similar results. See also Supplementary Fig. 7b, c for additional Co-IP data. Source data are provided as a Source Data file.

In line with this, we demonstrate that both human and *Xenopus leavis* CSB require its CIM to directly interact with CSA in vitro. However, the CIM is absent in species without CSA, including yeast, nematodes, but also holometabolous insects, which have lost the *CSA* gene during the course of evolution (Supplementary Fig. 5).

It is striking that even though CSB contains a CIM, the association between these proteins is induced by UV irradiation. Previous studies revealed that the association of CSB with lesion-stalled RNAPIIo triggers a conformational change that repositions the N-terminus, thereby exposing residues in the C-terminus of CSB[49]. It is conceivable that this conformational

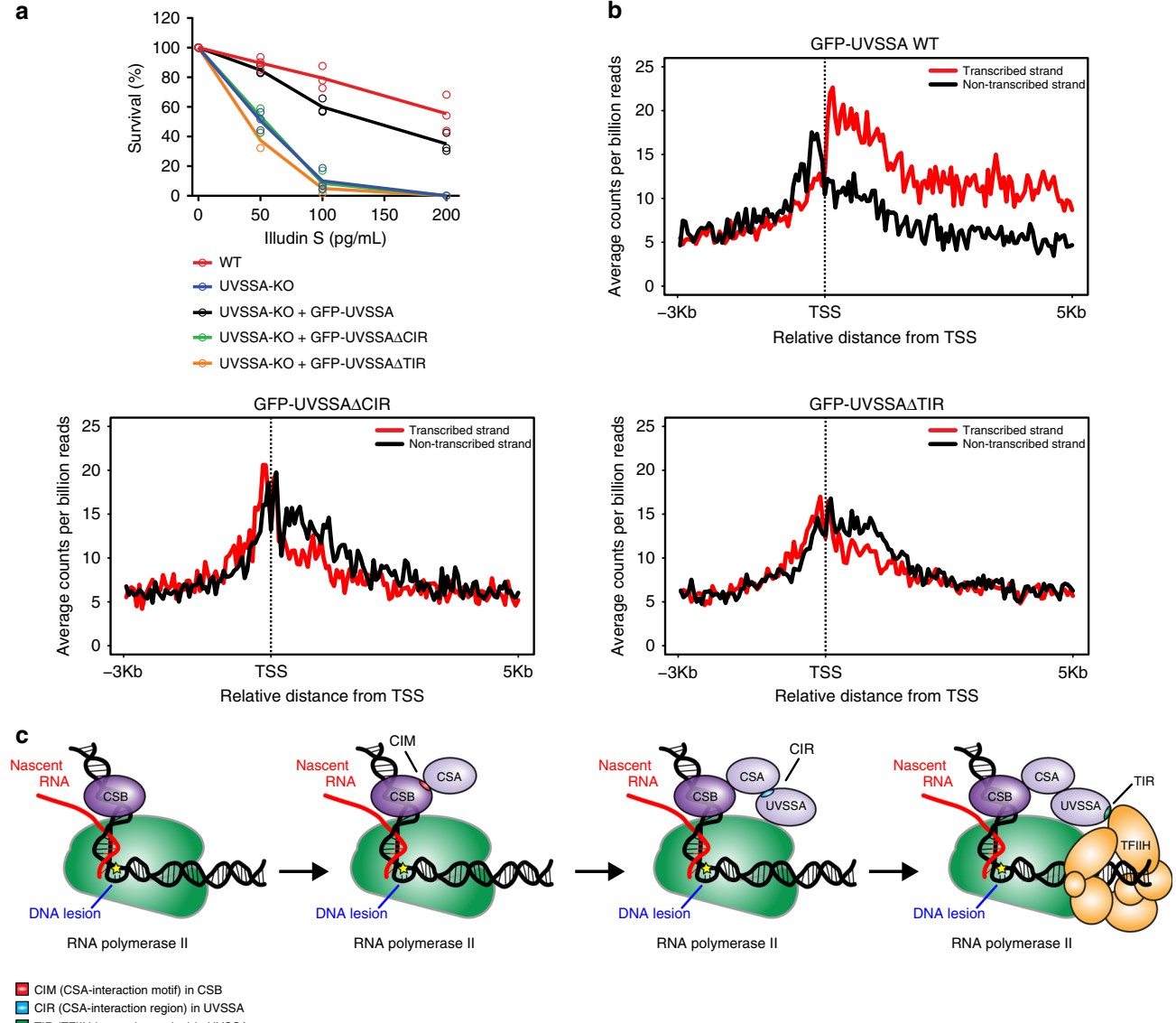

**Fig. 7 The CIR and TIR of UVSSA are crucial for TCR. a** Clonogenic Illudin S survival of WT and UVSSA-KO cell lines and the GFP-tagged UVSSA rescue cell lines. Each symbol represents the mean of an independent experiment ($n = 3$ for all except for GFP-UVSSA$^{\Delta CIR}$ which is $n = 2$), each of which is based on two technical replicates. Note that the same survival data for UVSSA-KO is also shown in Fig. 4b. **b** CPD XR-seq repair signal 3 Kb upstream and 5 Kb downstream of the annotated TSS of 16,088 genes in GFP-UVSSA$^{WT}$, GFP-UVSSA$^{\Delta CIR}$, and GFP-UVSSA$^{\Delta TIR}$. Signal is plotted separately for the transcribed (red) and non-transcribed (black) strands. The data represent the average of two independent experiments for GFP-UVSSA$^{WT}$ and one single experiment for GFP-UVSSA$^{\Delta CIR}$ and GFP-UVSSA$^{\Delta TIR}$ with a bin size of 53 nt. See also Supplementary Fig. 9 for additional XR-seq data. **c** Model of how the assembly of CSB, CSA, and UVSSA targets the TFIIH complex to DNA damage-stalled RNAPIIo. Source data are provided as a Source Data file.

change exposes the CIM to facilitate efficient CSA recruitment. Interestingly, while the CIM is located right next to the UBD in CSB[42], we find that CSB$^{\Delta UBD}$ is fully functional in interacting with CSA. However, it is possible that the CIM and the UBD collaborate, as a tandem protein-interaction module[50], to enable optimal CSA recruitment. In this scenario, CSA would have protein–protein interactions with the CIM, which would be stabilized by the binding of the UBD to auto-ubiquitylated CSA[27].

The recently identified UVSSA protein can be isolated as part of a chromatin-bound stalled RNAPIIo complex. Our current findings shed light on its recruitment mechanism by demonstrating that the association of UVSSA with RNAPIIo is strongly induced by UV irradiation and fully dependent on both CSA and CSB. Moreover, knockout of UVSSA did not affect CSA or CSB recruitment to DNA damage-stalled RNAPIIo, suggesting that UVSSA is the last of these proteins to be recruited. Consistent

with a reported association between CSA and UVSSA[32], we find that CSA targets UVSSA to DNA damage-stalled RNAPIIo by interacting with a region in the N-terminal VHS domain (CIR; amino acids 100–200) of UVSSA. Intriguingly, the robust UV-induced association between CSA and UVSSA is stabilized by CSB, suggesting a cooperative assembly mechanism of the TCR complex.

In contrast to our observation that the CS proteins are required for the recruitment of UVSSA to DNA damage-stalled RNAPIIo, live-cell imaging experiments showed that UVSSA is recruited to sites of UV-C-induced laser damage independently of the CS proteins[25,31]. There could be several reasons for these seemingly conflicting results. Firstly, the methodology is very different. We isolate RNAPIIo-associated TCR proteins from the chromatin-bound fraction after UV, while live-cell imaging studies monitor the recruitment of GFP-tagged TCR proteins to local UV-C laser

damage. Therefore, it is possible that the observed recruitment of CSB and UVSSA could, in part, be triggered by something other than DNA damage-stalled RNAPIIo. In line with this hypothesis, GFP-CSA could not be detected at sites of local UV-C laser damage[31], even though CSA is essential for TCR and showed a robust UV-specific association with RNAPIIo under our conditions. Secondly, the time frame during which UVSSA association is measured is different. While we isolate RNAPIIo-associated UVSSA 1 h after UV irradiation, the recruitment studies visualized UVSSA binding in the first 40 s after UV-C laser irradiation. It cannot be excluded that UVSSA transiently associates with UV-damaged chromatin independently of the CS proteins, but that the stable association with stalled RNAPIIo during productive TCR is fully dependent on CSA and CSB. In line with this, we find that mutants of TCR proteins that display a clear assembly defect under our conditions also show a strong sensitivity to Illudin S reflecting impaired TCR. In conclusion, our findings favor a model in which UVSSA is recruited by CSA and argues for a cooperative assembly mechanism in which CSB stabilizes the association between CSA and UVSSA to ensure efficient targeting to stalled RNAPIIo.

A major unresolved question is how the core NER machinery, likely starting with the TFIIH complex, is recruited to DNA damage-stalled RNAPIIo to initiate repair. In vitro experiments have shown that the association of CSB with RNAPII is sufficient to recruit TFIIH[23]. In addition, CSA was shown to associate with the p44 subunit of TFIIH[26], while UVSSA can interact with the p62 subunit of TFIIH[44]. In agreement, we found that GFP-UVSSA associates with several subunits of the TFIIH complex in a UV-specific manner in vivo. Furthermore, our data reveals that CSB, CSA, and UVSSA are equally important for the recruitment of TFIIH to DNA damage-stalled RNAPIIo in vivo. Indeed, similar to previous results with CSB-deficient cells[46,51], our high-resolution repair maps fully support a crucial role of both CSA and UVSSA in the TCR-mediated clearing of UV-induced lesions on a genome-wide level. Importantly, we found that UVSSA contains a TFIIH-interacting region (TIR; amino acids 400–500), which is crucial for the association of TFIIH with DNA damage-stalled RNAPIIo. Consistently, it has been shown that the PH domain of p62 (1–108) associates with a small fragment in UVSSA (400–419) in vitro and that mutations within this region causes a defect in recovery of RNA synthesis in vivo[44]. Moreover, we found that the UVSSA$^{\Delta CIR}$ mutant was not only unable to associate with CSA, but also with the TFIIH complex. Our findings favor a model in which CSA not only recruits UVSSA to stalled RNAPIIo but also stabilizes the direct interaction between UVSSA and TFIIH, resulting in the recruitment of TFIIH to DNA damage-stalled RNAPIIo. In this regard, it would be interesting to examine if this interaction between UVSSA and the p62 subunit of TFIIH is the sole mechanism through which TFIIH is recruited to DNA damage-stalled RNAPIIo in vivo, or whether other subunits and regions also contribute.

Here we show that UVSSA is essential to bridge the TFIIH complex to CSB/CSA-bound RNAPIIo to initiate TCR. Importantly, these findings also suggest that neurodegeneration seen in CS is not caused by the inability to remove transcription-blocking DNA lesions, since neurodegeneration is not a feature in UV$^S$S. Previous findings revealed that CS fibroblasts fail to ubiquitylate and subsequently degrade DNA damage-stalled RNAPIIo[16], while UV$^S$S fibroblasts displayed even faster degradation of RNAPIIo after UV, possibly due to a failure to deubiquitylate RNAPIIo by the UVSSA binding partner USP7[16,17,25,32]. These findings suggest that features of CS are caused by toxicity associated with prolonged RNAPIIo stalling at DNA lesions rather than the inability to remove transcription-blocking DNA lesions[52].

It remains unresolved how RNAPII is repositioned while TCR is in progress to enable access to the DNA lesion. We speculate that the TCR complex that we capture contains an inactive TFIIH complex, which is bound to the CAK complex. Indeed, we detect the association of CAK subunit CDK7 with RNAPIIo after UV. The association of repair factors XPA and XPG activates TFIIH[34,35], which could cause the backtracking, or even removal of RNAPII. This model would explain why we do not detect the association of downstream repair factors with DNA damage-stalled RNAPII.

We propose a model in which CSB is the first protein to be recruited to DNA damage-stalled RNAPIIo (Fig. 7c). This binding of CSB could bring about a conformational change, thereby exposing the newly identified CIM to facilitate efficient CSA recruitment through direct protein–protein contacts. Once bound, CSA targets UVSSA to DNA damage-stalled RNAPIIo, and this interaction is stabilized by CSB. UVSSA, in turn, mediates the recruitment of the TFIIH complex in a cooperative manner that is stabilized by both CSB and CSA. Although both CS proteins could interact with TFIIH, it is likely that only CSA contributes directly to this stabilization, while CSB contributes indirectly through ensuring the association of CSA itself and stabilizing the interaction between CSA and UVSSA. At the stage when TFIIH is bound, it seems likely that RNAPIIo and CSB/CSA/UVSSA are displaced and that the TCR-specific pre-incision complex is assembled starting with XPA. It is interesting to note that the yeast orthologue of CSB, RAD26, is bound to the DNA upstream of RNAPII[53], while human TFIIH in the transcription pre-initiation complex is bound downstream of RNAPII[54]. If TFIIH is recruited to the same side of RNAPII during TCR, it suggests that CSB/CSA/UVSSA extend from the upstream to the downstream DNA around RNAPII to position TFIIH. It will be very interesting to gain structural insights into these molecular events. In conclusion, our findings reveal the recruitment mechanism of the TFIIH complex to DNA damage-stalled RNAPII, which involves the sequential and cooperative assembly of the CSB, CSA, and UVSSA proteins.

## Methods

**Cell lines**. Cell lines (listed in Supplementary Table 1) were cultured at 37 °C in an atmosphere of 5% CO$_2$ in DMEM (Thermo Fisher Scientific) supplemented with penicillin/streptomycin (Sigma) and 10% Fetal bovine serum (FBS; Bodinco BV). Sf9 cells were cultured in ESF 921 insect cell culture medium (Fisher Scientific). U2OS 2-6-3 cells containing 200 copies of a LacO-containing cassette (~4 Mbp) were a gift from Susan Janicki[39]. UVSSA-deficient KPS3-hTERT cells and their UVSSA-rescued counterparts were a gift from Tomoo Ogi[16]. U2OS Flp-In/T-REx cells, which were generated using the Flp-In/T-REx system (Thermo Fisher Scientific), were a gift from Daniel Durocher[50].

**Generation of knockout cell lines**. To generate stable knockouts, U2OS Flp-In/T-REx cells were co-transfected with pLV-U6g-PPB encoding a guide RNA from the LUMC/Sigma-Aldrich sgRNA library (see Supplementary Table 2 for plasmids, Supplementary Table 3 for sgRNA sequences) together with an expression vector encoding Cas9-2A-GFP (pX458; Addgene #48138) using lipofectamine 2000 (Invitrogen). Transfected cells were selected on puromycin (1 µg/mL) for 3 days, plated at low density after which individual clones were isolated. To generate double knockouts, single knockout clones were transfected with pLV-U6g-PPB encoding a sgRNA together with pX458 encoding Cas9, cells were FACS sorted on BFP/GFP, plated at low density after which individual clones were isolated. Isolated knockout clones were verified by western blot analysis and/or sanger sequencing. The absence of Cas9 integration/stable expression was confirmed by western blot analysis.

**PCR analysis of knockout clones**. Genomic DNA was isolated by resuspending cell pellets in WCE buffer (50 mM KCl, 10 mM Tris pH 8.0, 25 mM MgCl$_2$ 0.1 mg/mL gelatin, 0.45% Tween-20, 0.45% NP-40) containing 0,1 mg/mL Proteinase K (EO0491;Thermo Fisher Scientific) and incubating for 1 h at 56 °C followed by a 10 min heat inactivation of Proteinase K by 96 °C. Fragments of ~1 kb, containing the sgRNA sequence, were amplified by PCR (sequencing primers are listed in Supplementary Table 4) followed by Sanger sequencing using either the forward or the reverse primer.

**Generation of stable cell lines**. Selected knockout clones of CSB, CSA, and UVSSA (see Supplementary Table 1) were subsequently used to stably express GFP-CSB$^{WT}$, GFP-CSB$^{\Delta CIM}$, CSA$^{WT}$-GFP, GFP-UVSSA$^{WT}$, GFP-UVSSA$^{\Delta CIR}$, and GFP-UVSSA$^{\Delta TIR}$ by co-transfection of pCDNA5/FRT/TO-Puro plasmid encoding these CSB, CSA, and UVSSA variants (2 μg), together with pOG44 plasmid encoding the Flp recombinase (0.5 μg). After selection on 1 μg/mL puromycin and 4 μg/mL blasticidin S, single clones were isolated and expanded. Clones were selected based on their near-endogenous expression level compared with parental U2OS Flp-In/T-REx cells. Expression of these GFP-tagged TCR proteins was induced by the addition of 2 μg/mL doxycycline for 24 h.

**Plasmid constructs**. The Neomycin resistance gene in pcDNA5/FRT/TO-Neo (Addgene #41000) was replaced with a Puromycin resistance gene. Fragments spanning GFP-N1 (clontech) and GFP-C1 (clontech) including the multiple cloning site were inserted into pcDNA5/FRT/TO-puro. CSB$^{WT}$, CSA$^{WT}$, and UVSSA$^{WT}$ were amplified by PCR (see Supplementary Table 5 for primers) and inserted into pcDNA5/FRT/TO-Puro-GFP-N1 or pcDNA5/FRT/TO-Puro-GFP-C1 and in mCherry-LacR-NLS-C1/C3. Deletion constructs of CSB and UVSSA were generated by site-directed mutagenesis PCR. All sequences were verified by sequencing.

**Illudin S survival assay**. Knockout and rescue cell lines were trypsinized, seeded at low density, and mock-treated or exposed to a dilution series of Illudin S (Santa cruz; sc-391575) for 72 h (30, 60, 100 pg/mL or 50, 100, and 200 pg/mL). On day 10, the cells were washed with 0.9% NaCl and stained with methylene blue. Colonies of more than 20 cells were scored.

**Immunoprecipitation for Co-IP**. Cells were irradiated with UV-C light (20 J/m$^2$) or mock treated and harvested 1 h after UV. Chromatin-enriched fractions were prepared by incubating the cells for 20 min on ice in IP buffer (IP-130 for endogenous RNAPII IP and IP-150 for GFP-IP), followed by centrifugation, and removal of the supernatant. For Co-IP of endogenous RNAPIIo, the chromatin-enriched cell pellets were lysed in IP-130 buffer (30 mM Tris pH 7.5, 130 mM NaCl, 2 mM MgCl$_2$, 0.5% Triton X-100, protease inhibitor cocktail (Roche), 250 U/mL Benzonase Nuclease (Novagen)), and 2 μg RNAPII-S2 (ab5095, Abcam) for 2–3 h at 4 °C. For GFP IPs, the chromatin-enriched cell pellets were lysed in IP-150 buffer (50 mM Tris pH 7.5, 150 mM NaCl, 0.5% NP-40, 2 mM MgCl$_2$, protease inhibitor cocktail (Roche), and 500 U/mL Benzonase Nuclease (Novagen)) for 1 h at 4 °C. Protein complexes were pulled down by 1.5 h incubation with Protein A agarose beads (Millipore) or GFP-Trap A beads (Chromotek). For subsequent analysis by western blotting, the beads were washed six times with IP-130 buffer for endogenous RNAPII IP and EBC-2 buffer (50 mM Tris pH 7.5, 150 mM NaCl, 1 mM EDTA, 0.5% NP-40, and protease inhibitor cocktail (Roche)) for GFP-IPs. The samples were prepared by boiling in Laemmli-SDS sample buffer. Unless indicated otherwise, all IP experiments were performed on the chromatin fraction. At least two independent replicates of each IP experiment were performed.

**Generation of mass spectrometry samples**. For the generation of mass spectrometry samples, the beads were washed four times with EBC-2 buffer without NP-40 and two times with 50 mM ammonium bicarbonate followed by overnight digestion using 2.5 μg trypsin at 37 °C under constant shaking. The bead suspension was loaded onto a 0.45 μm filter column (Millipore) to elute the peptides. The peptides were passed through a C-18 stage tips for desalting. The stagetips were activated by washing with methanol followed by washing with buffer B (80% acetonitrile and 0.1% formic acid) and 0.1% formic acid. Peptides were acidified with 2% trifluoroacetic acid and loaded on the stagetips. The peptides were eluted twice with 25 μl 60% acetonitrile/0.1% formic acid and lyophilized. At least three biological repeats for each condition were performed.

**Mass spectrometry**. Mass spectrometry analysis was performed on a Q-Exactive Orbitrap mass spectrometer (Thermo Scientific, Germany) coupled to an EASY-nanoLC 1000 system (Proxeon, Odense, Denmark). The mass spectrometer was operated in positive-ion mode at 2.9 kV with the capillary heated to 250 °C in a data-dependent acquisition mode with a top 7 method, Precursor ions with a charge state of one and greater than six were excluded from triggering MS/MS events. For the UV-dependent UVSSA interactors (Fig. 5b), samples were analyzed essentially as previously described[55]. Digested peptides were separated using a 15 cm fused silica capillary (ID: 75 μm; OD: 375 μm; Polymicro Technologies, California, US) in-house packed with 1.9 μm C18-AQ beads (Reprospher-DE, Pur, Dr. Maisch, Ammerburch-Entringen, Germany). Peptides were separated by liquid chromatography using a gradient from 2% to 95% acetonitrile with 0.1% formic acid at a flow rate of 200 nl/min for 65 min. Full scan MS spectra were obtained with a resolution of 70,000, a target value of $3 \times 10^6$, and a scan range from 400 to 2000 m/z. Maximum injection time (IT) was set to 50 ms. Higher-collisional dissociation (HCD) tandem mass spectra (MS/MS) were recorded with a resolution of 35,000, a maximum IT of 120 ms, a target value of $1 \times 10^5$ and a normalized collision energy of 25%. Minimum AGC target was set to $10^3$. The precursor ion masses selected for MS/MS analysis were subsequently dynamically excluded from MS/MS analysis for 60 s. For interactors of the CSB and UVSSA mutant proteins

(Supplementary Figs. 6a–c and 8a–e), mass spectrometry analysis was performed as previously described[55]. Peptides were separated in a pre-cut 25 cm silica emitter (FS360-75-15-N-5-C25, MS Wil B.V, The Netherlands), in-house packed with 1.9 μm C18-AQ beads (Reprospher-DE, Pur, Dr. Maisch, Ammerburch-Entringen, Germany). Peptides were separated by liquid chromatography using a gradient from 2 to 95% acetonitrile with 0.1% formic acid at a flow rate of 200 nl/min for 125 min. Full scan MS spectra were obtained with a resolution of 70,000, a target value of $3 \times 10^6$, and a scan range from 300 to 1600 m/z. Maximum IT was set to 250 ms. HCD tandem mass spectra (MS/MS) were recorded with a resolution of 35,000, a maximum IT of 120 ms, a target value of $1 \times 10^5$ and a normalized collision energy of 25%. Minimum AGC target was set to $10^4$. Dynamic exclusion was set to 40 s.

**Mass spectrometry data analysis**. Raw mass spectrometry files were analyzed with MaxQuant software (version 1.5.3.30) as described[56], with the following modifications from default settings: the maximum number of mis-cleavages by trypsin was set to 4, Label Free Quantification (LFQ) was enabled thereby disabling the Fast LFQ feature. Match-between-runs feature was enabled with a match time window of 0.7 minutes and an alignment time window of 20 min. We performed the search against an in silico digested UniProt reference proteome for Homo sapiens (14th December 2017). Analysis output from MaxQuant was further processed in the Perseus (version 1.5.5.3) computational platform[57]. Proteins identified as common contaminants, only identified by site and reverse peptide were filtered out, and then all the LFQ intensities were log2 transformed. Different biological repeats of each condition were grouped and only protein groups identified in all biological replicates in at least one condition were included for further analysis. Missing values were imputed using Perseus software by normally distributed values with a 1.8 downshift (log2) and a randomized 0.3 width (log2) considering total matrix values. Volcano plots were generated and Student's $t$ tests were performed to compare the different conditions. Spreadsheets from the statistical analysis output from Perseus were further processed in Microsoft Excel for comprehensive visualization and analysis of the data. A web app (VolcaNoseR) was made with R/Shiny for generating and sharing interactive volcano plots. Links to the interactive volcano plots are listed in Supplementary Table 7 and in the relevant figure legends.

**Western blot**. Proteins were separated on 4–12% Criterion XT Bis-Tris gels (Bio-Rad, #3450124) in NuPAGE MOPS running buffer (NP0001-02 Thermo Fisher Scientific), and blotted onto PVDF membranes (IPFL00010, EMD Millipore). The membrane was blocked with blocking buffer (Rockland, MB-070-003) for 2 h at RT. The membrane was then probed with antibodies (listed in Supplementary Table 6) as indicated.

**Slot blot for CPDs after IP**. Cells were irradiated with UV-C light (20 J/m$^2$) or mock treated and crosslinked with 0.5 mg/mL disuccinimidyl glutarate (DSG; Thermo Fisher) in PBS for 45 min at room temperature. Cells were washed with PBS and crosslinked with 1% PFA for 20 min at room temperature. Fixation was stopped by adding 1.25 M glycine in PBS to a final concentration of 0.1 M for 3 min at room temperature. Cells were washed with cold PBS and lysed and collected in a buffer containing 0.25% Triton X-100, 10 mM EDTA (pH 8.0), 0.5 mM EGTA (pH 8.0), and 20 mM Hepes (pH 7.6). Chromatin was pelleted in 5 min at 400 g and incubated in a buffer containing 150 mM NaCl, 1 mM EDTA (pH 8.0), 0.5 mM EGTA (pH 8.0), and 50 mM Hepes (pH 7.6) for 10 min at 4 °C. Chromatin was again pelleted for 5 min at 400 g and resuspended in ChIP-buffer (0.15 % SDS, 1% Triton X-100, 150 mM NaCl, 1 mM EDTA (pH 8.0), 0.5 mM EGTA (pH 8.0), and 20 mM Hepes (pH 7.6)) to a final concentration of $20 \times 10^6$ cells/mL. Chromatin was sonicated to approximately one nucleosome length using a Bioruptor waterbath sonicator (Diagenode). Chromatin of $20 \times 10^6$ cells was incubated with 6 μg RNAPII-S2 (ab5095, Abcam) overnight at 4 °C, followed by a 1.5 h protein-chromatin pull-down with a 1:1 mix of protein A and protein G Dynabeads (Thermo Fisher; 10001D and 10003D). The beads were washed extensively, followed by decrosslinking for 4 h at 65 °C in the presence of proteinase K and RNAse A. The DNA was purified and concentrated using MinElute columns (Qiagen) and the concentration was measured using a dsDNA HS Qubit assay (Thermo Fisher Scientific). 15 ng of DNA for each sample was denatured for 15 min at 98 °C and blotted on Hybond N+ (RPN203B, GE healthcare). The membrane was dried for at least 1 h at 57 °C and blocked overnight in 5% milk in PBS + 0.1% tween. CPDs were detected using anti-CPD mouse monoclonal antibody followed by incubation with an anti-mouse HRP conjugated antibody (Supplementary Table 6). CPD signals were detected using an ECL reagent (Sigma, GERPN2232).

**Chromatin tethering**. U2OS 2-6-3 cells containing 200 copies of a LacO-containing cassette were co-transfected with lipofectamine 2000 (Invitrogen) and plasmid DNA for 6 h at 37 °C in an atmosphere of 5% CO$_2$. 24 h after transfection the cells were fixed with 4% paraformaldehyde (Sigma) in PBS for 15 min. The cells were either permeabilized with 0.5% triton X-100 (Sigma) in PBS for 10 min and mounted in poly mount (Polysciences; 18606) or subjected to immunofluorescent labeling.

**Immunofluorescent labeling**. Cells were permeabilized with 0.5% triton X-100 (Sigma) in PBS for 10 min, followed by treatment with 100 mM glycine in PBS for 10 min to block unreacted aldehyde groups. Cells were rinsed with PBS and equilibrated in wash buffer (WB: PBS containing 0.5% BSA, and 0.05% Tween-20 (Sigma-Aldrich)) for 10 min. Antibody steps and washes were in WB. The primary antibody rabbit-p89 (1/100; Santa Cruz; SC-293; S19) was incubated for 2 h at RT. Detection was done using goat-rabbit Ig coupled to Alexa 488 (1:1000; Invitrogen). Cells were incubated with 0.1 μg/mL DAPI and mounted in Poly mount (Poly-sciences; 18606).

**Microscopic analysis of fixed cells**. Images of fixed samples were acquired on a Zeiss AxioImager M2 or D2 widefield fluorescence microscope equipped with a 63× PLAN APO (1.4 NA) oil-immersion objectives (Zeiss) and an HXP 120 metal-halide lamp used for excitation. Fluorescent probes were detected using the following filters: DAPI (excitation filter: 350/50 nm, dichroic mirror: 400 nm, emission filter: 460/50 nm), GFP/Alexa 488 (excitation filter: 470/40 nm, dichroic mirror: 495 nm, emission filter: 525/50 nm), mCherry (excitation filter: 560/40 nm, dichroic mirror: 585 nm, emission filter: 630/75 nm). Images were recorded using ZEN 2012 software (blue edition, version 1.1.0.0).

**Genome-wide XR-sequencing**. XR-seq was performed as previously described[46,51]. Briefly, cells were harvested 3 h after treatment with 20 J/m$^2$ UVC (254 nm). Primary excision products were pulled down by TFIIH coimmunopre-cipitation with anti-p62 and anti-p89 antibodies (Santa Cruz Biotechnology sc25329 and sc271500), and ligated to both 5′ and 3′ adapters. Ligation products containing CPD were purified by immunoprecipitation with the anti-CPD anti-body (Cosmo Bio NM-DND-001) and repaired in vitro by Drosophila melano-gaster CPD photolyase. Repaired DNA was PCR-amplified with Index primers and purified by 10% native polyacrylamide gels. Libraries were pooled and sequenced either on a HiSeq 2500 or on a NextSeq 500 to produce at least 4 million reads per sample. Quality score for each nucleotide was analyzed using the Fastx-toolkit to ensure only high-quality reads are processed. Adapter sequence was trimmed from each read using Trimmomatic (version 0.36)[58]. Reads were aligned to the genome using Bowtie (version 1.2.2)[59]. Following alignment, reads that were mapped to chromosome Y or mitochondrial chromosome were filtered (U2OS cell line is derived from female bone tissue) and PCR duplicates were removed using PicardCommandLine MarkDuplicates (version 2.8.1) (http://broadinstitute.github.io/picard/). There were high levels of PCR duplicates due to low efficiency of excised oligo recovery, but these were sufficient for analysis of TCR. To plot average XR-seq signal along genes, the genes annotation file was downloaded from Ensembl, assembly GRCh38, release 96. Non-overlapping regions around the TSS were obtained using custom scripts and BEDTools (version 2.26.0) slop and merge commands[60]. All samples were converted to BED format using bedtools bamtobed command. Strand-specific profiles over the TSS were created using the R (version 3.5.1) Bioconductor genomation package (version 1.14.0)[61]. For U2OS (FRT) WT, UVSSA-KO, and GFP-UVSSA the experiment was done in duplicate. For CSA-KO, GFP-UVSSA$^{\Delta CIR}$ and GFP-UVSSA$^{\Delta TIR}$ the experiment was done once.

**Protein expression and purification**. Coding sequences of *Xenopus laevis* CSB and CSA-DDB1-CUL4-RBX1 (CRL4$^{CSA}$), as well as human CSB were amplified from cDNA clones or ordered as codon-optimized gene blocks from Integrated DNA Technologies. All open reading frames were cloned into pAceBac1 (pAB1) or pIDC vectors containing the indicated affinity tags (Supplementary Table 2). For the generation of CRL4$^{CSA}$, CSA/DDB1, and CUL4A/RBX1 heterodimers were cloned into separate vectors, respectively. To obtain bacmids for insect cell expression, plasmids were transformed into chemically competent DH10Bac cells and purified using ZR BAC DNA miniprep kit (Zymo Research). Baculoviruses encoding CSB variants, CSA/DDB1, or CUL4A/RBX1 were amplified in three stages (P1, P2, and P3) in Sf9 cells (Expression Systems). Protein expression was performed for 72 h in 500 mL Sf9 cells per construct infected with 10 mL P2 or P3 baculovirus. Cells were cultured at 27 °C in ESF 921 insect cell culture medium (Fisher Scientific), pelleted at 1000 × g for 15 min, frozen in liquid nitrogen, and stored at −80 °C. Protein purifications were performed at 4 °C. Cell pellets were resuspended in a final volume of 50 mL wash buffer (50 mM HEPES [pH 7.5], 300 mM NaCl, 10% glycerol) containing 0.1% NP-40 and one EDTA-free cOmplete protease inhibitor tablet (Roche). Cells were lysed by sonication and cleared by centrifugation for 1 h at 30,000 × g. The clarified lysate was incubated with 0.3–0.6 mL pre-equilibrated anti-FLAG M2 Affinity Gel (Sigma) for 1 h at 4 °C on a rotating wheel. The resin was washed extensively with Wash Buffer, and proteins were eluted with wash buffer containing 0.2 mg/mL 3xFLAG peptide (Sigma). CSB proteins were further purified by gel filtration (Superdex 200 Increase) in wash buffer containing 2 mM DTT, and pooled peak fractions were concentrated with 5 mL 10 MWCO spin concentrators (Millipore), frozen in liquid nitrogen, and stored at −80 °C. Eluted CSA-StrepII/FLAG-DDB1 complex was applied to 0.3 mL pre-equilibrated Strep-Tactin XT Superflow high capacity resin in a disposable gravity-flow column and washed 5x with 0.6 mL Wash Buffer. FLAG peptide-eluted FLAG-CUL4A/RBX1 complex was incubated with the immobilized CSA-StrepII/FLAG-DDB1 complex for 1 h at 4 °C to assemble CRL4$^{CSA}$. The resin was washed 5× with 0.6 mL wash buffer to remove excess FLAG-CUL4A/RBX1, and

CRL4$^{CSA}$ was eluted with BXT Buffer (iba-lifesciences), which contains 50 mM biotin. Pooled fractions were dialyzed O/N into 0.5× wash buffer containing 2 mM DTT, concentrated with 0.5 mL 3 MWCO spin concentrators (Millipore), frozen in liquid nitrogen, and stored at −80 °C.

**Pull-down using immobilized CSB proteins**. Purified FLAG-tagged CSB proteins were immobilized on pre-equilibrated anti-FLAG M2 Magnetic Beads (Sigma) for 2 h at 4 °C. The beads were washed 3× with 0.3 ml Pull-down Buffer (20 mM HEPES [pH 7.5], 100 mM KCl, 5 mM MgCl$_2$, 0.5 mM EDTA, 0.25 mg/mL BSA, 0.03% Tween) and incubated with *Xenopus laevis* nucleoplasmic extract (NPE) for 1 h at 4 °C. The beads were washed 3× with 0.3 ml Pull-down Buffer and resus-pended in Laemmli-SDS sample buffer. Samples were resolved by SDS-PAGE and analyzed by western blot.

**In vitro ubiquitylation assay**. Purified xlCRL4$^{CSA}$ was neddylated in vitro using the NEDD8 Conjugation Initiation Kit (Boston Biochem) according to the manu-facturer's protocols, except using 0.5x Uba3, 0.5x UbcH12, and 0.33x NEDD8 as compared with the recommended final concentrations. The reaction was incubated for 25 min at RT immediately prior to the in vitro ubiquitylation reaction, which contained the following final concentrations in Ubiquitylation Buffer (40 mM Tris pH 7.5, 10 mM MgCl$_2$, 0.6 mM DTT): 100 nM E1 (Enzo Life Sciences), 2.5 μM UBE2D2 (Boston Biochem), ~50 nM neddylated xlCRL4$^{CSA}$, 50 μM ubiquitin, 10 mM ATP, and 200–250 nM CSB protein. Reaction were incubated for indicated times at RT and stopped in Laemmli-SDS sample buffer prior to SDS-PAGE and western blot analysis.

**Reporting summary**. Further information on research design is available in the Nature Research Reporting Summary linked to this article.

## Data availability

Mass spectrometry proteomics data are presented in main Fig. 5b and Supplementary Figs. 6a–c and 8a–e, and have been deposited to the ProteomeXchange Consortium via the PRIDE partner repository[62] (https://www.ebi.ac.uk/pride/) with the dataset identifiers PXD013572 and PXD017329. Processed mass spectrometry proteomics data used for interactive Volcano plots is deposited on Zenodo.org (https://doi.org/10.5281/zenodo.3713174). The code for VolcaNoseR (Version 1.0.0) has been deposited on Zenodo.org (https://doi.org/10.5281/zenodo.3625858). XR-seq data are presented in main Figs. 5f, 7b, and Supplementary Figs. S7a, S9a. XR-seq data are deposited in the Gene Expression Omnibus (GEO; https://www.ncbi.nlm.nih.gov/geo/) under GSE132840. Additional data and custom code will be made available upon reasonable request. The source data underlying Figs. 1b, d–h, 2b, d–h, 3b–g, 4a–d, 5a, c–e, 6c–f, and 7a and Supplementary Figs. 1a–c, 2c, 3c, 4c, 6d–e, and 7b–c are provided as a Source Data file.

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

## Acknowledgements

The authors acknowledge Jean-Marc Egly and Rick Wood for their generous gift of TFIIH and XPA antibodies, respectively, and the genomics core research facility at the faculty of medicine in the Hebrew University for technical advice. Tom Misteli provided LacR-NLS plasmid, Tomoo Ogi provided UVSSA-deficient KPS3-hTERT cells, Susan Janicki provided U2OS 2-6-3 cells. This work was funded by an LUMC Research Fellowship and an NWO-VIDI grant (ALW.016.161.320) to M.S.L., an ERC starting grant

(310913) to A.C.O.V., a Young Investigator Grant from the Dutch Cancer Society (KWF-YIG: 11367) to R.G-P., a Manja Gideon Foundation scholarship to H.G.-B., and Israel Science Foundation grants (1710/17 and 1762/17) administered by the Israeli Academy for Science and humanities and The Israel Cancer Association grant (20191630) to S.A. S.A. is the recipient of the Jacob and Lena Joels memorial fund senior lectureship. J.C.W. was supported by NIH grant HL098316 and is a Howard Hughes Medical Institute (HHMI) Investigator and an American Cancer Society Research Professor. T.E.T.M. was supported by an EMBO Long-term fellowship (ALTF 1316-2016) and an HHMI fellowship of The Jane Coffin Childs Memorial Fund for Medical Research.

## Author contributions

Y.v.d.W. generated knockout cells, constructs and stable cell-lines, performed LacR-based tethering assays, clonogenic survivals, PCR and western blot analysis to validate knockouts, Co-IP experiments for western blot analysis, Co-IP experiments for mass spectrometry, slot blot assays, and wrote the paper. K.A. generated stable cell-lines, western blot analysis to validate knockouts, and Co-IP experiments. R.G.-P. and A.C.O.V. analyzed the mass spectrometry samples. H.G.-B. performed XR-seq. H.G.-B., E.E.H., and S.A. analyzed the XR-seq samples. J.G. developed VolcaNoseR. T.E.T.M. generated recombinant CSB proteins and xlCRL4$^{CSA}$, and performed pull-down and in vitro ubiquitylation assays. D.v.d.H. generated knockout cells, constructs, and performed western blot analysis to validate knockouts, and Co-IP experiments. J.C.W. supervised T.E.T.M. M.S.L. supervised the project and wrote the paper.

## Competing interests

The authors declare no competing interests.
