## [Peer Review File · Nature Communications]

Reviewers' comments:

Reviewer #1 (Remarks to the Author):

The Luijsterburg group has a longstanding interest in the understanding of DNA repair mechanism. Here he described a sequential recruitment of CSB, CSA, UVSSA, TFIIH on the stalled RNA polymerase in front of a lesion, the cells being previously UV irradiated. He also map interactions between the different partners. However the functional result of this complex is missing since several enzymes such as XPB, CSA/CUL4, CDK? are there. Could you comment? Despite the well performed analysis of the formation of such complex named TCR, I have several concerns.

1). According to their data, there is a large amount of Pol II stalled in front of a lesion. Could you discriminate between elongating Pol II and stalled Pol II since all the genes might not be damaged and not all Pol II blocked?

Why the immuno-fluorescence technique established by Volker et al could not locate any Pol II /CSB while you IP a large amount of Pol II and CSB?

2). Why immuno-precipitations using CPD antibodies were not performed? ChIP/re ChIP investigation might demonstrate that we have stalled Pol II as a good control.

3). You demonstrate that these factors are sequentially recruited. Control experiments using antibodies towards these proteins would be useful to really establish the stalled complex.

4) In your mass spectra experiments you show that a large amount of proteins co-IP with UVSSA. Why there is no the subunits of TFIIH such as the cdk7 or Cyclin H? What about the presence of p-Tefb that phosphorylates Ser 2 of Pol II-CTD?

5). Few years ago, it was demonstrated that TFIIH was released from Pol II once 50 nt were synthesized. In your model, TFIIH come back? Could you comment the CAK presence? In the Ebright and Coulombe models, TFIIH is located in front of Pol II. How its binding might occur in the case of stalled Pol II? Look at Cramer (Pol II in front of DNA lesion) and Nogales paper (Transcription complex).

6). What about the CUL4 and Roc protein that might ubiquitinate Pol II?

7). In your paper, you did not comment or mention papers from Proietti (EMBO, 2006) and Epantchisev (Mol Cell, 2017). How these findings fit (if any) in your model?

Minor points

- . Could you precise the role of illudin?
- . Fig 1f why no GFP-CSB in the input?
- . Some figures should be reduced: Fig 1a;

Reviewer #2 (Remarks to the Author):

Transcription-coupled repair (TCR) is a sub-pathway of the nucleotide excision repair (NER) that removes bulky lesions, for example generated by UV light, in the transcribed regions of the genome. Although proteins that function in NER are known, their assembly mechanism at sites of DNA lesions remains poorly understood.

In this manuscript, van der Weegen et al. report an analysis of the TCR protein assembly to stalled RNA pol 2 in a series of isogenic U2OS cell lines. Experiments in this manuscript are well controlled and carefully conducted, and the findings provide a relevant step forward in mechanistic understanding of TC-NER. The authors focused on the core TC-NER proteins CSA, CSB, UVSSA and TF2H. They performed a series of IPs in different TC-NER isogenic cell lines to follow the assembly of these proteins to stalled RNA pol 2 using western blotting. The main drawback of this study is the limited number of proteins the authors could probe for by WB. The study would have benefited greatly if the assembly of different complexes would have been followed by quantitative MS. This would make the study more comprehensive and interesting to the wider spectrum of readers. A second major issue is that XR-seq was performed in KO cells and not in cells reconstituted with mutants that the authors identified in the study. It is not surprising that CSA and UVSSA KO impair

NER. If authors expand on these aspects by performing quantitative interactomics and XR-seq to make the study more comprehensive, I support the publication in Nature Communications.

Main comments:

- The approach used in Figure 1 to study protein complexes assembled at UV-stalled RNA pol2 should be combined with quantitative MS to gain a more comprehensive view of recruited proteins. The authors state that the IP was done with an antibody against phospho-Ser2 to follow only proteins assembled at elongating and not initiating POL2. However, there is no data presented to show that these proteins are indeed recruited specifically to the elongating POL2 after UV. The authors should use MS to compare the proteins recruited to phospho-ser5 and ser2 after UV irradiation of cells.
- The authors should test how the interaction profile of different mutants changes after UV using quantitative MS. In this regard, the authors could pull down GFP-tagged CSB deltaCIM, CSA deltaCIR and UVSSA deltaTIR from respective isogenic lines and compare them to GFP-tagged WT proteins.
- The same mutants should be also tested in XR-seq in order to define the importance of identified interaction surfaces in TC-NER
- Can the authors show with recombinant purified proteins in in vitro assays the importance of identified CSA CIR and UVSSA TIR region for interaction between CSA, UVSSA and TF2H? Have the authors checked with purified proteins that CSB does not bind to UVSSA?

Minor comments:

- Scheme in Figure 1A is confusing and not understandable without reading the manuscript text. Why is the antibody depicted before IP? This should be made easier to follow for the reader.
- Why are there no error bars in some lines of the clonogenic cell survival in Figures 1D, 3E, 4B?
- The role of TF2H in TC-NER should be better explained in the introduction
- It is not clear what the Supplementary Figure 1 shows. It seems redundant to Figure 1. Are these the replicates of the same experiment?

Reviewer #1 (Remarks to the Author):

The Luijsterburg group has a longstanding interest in the understanding of DNA repair mechanism. Here he described a sequential recruitment of CSB, CSA, UVSSA, TFIIH on the stalled RNA polymerase in front of a lesion, the cells being previously UV irradiated. He also map interactions between the different partners. However the functional result of this complex is missing since several enzymes such as XPB, CSA/CUL4, CDK? are there. Could you comment?

Despite the well performed analysis of the formation of such complex named TCR, I have several concerns.

1) According to their data, there is a large amount of Pol II stalled in front of a lesion. Could you discriminate between elongating Pol II and stalled Pol II since all the genes might not be damaged and not all Pol II blocked?

We fully agree with the reviewer that not all RNAPII that we immunoprecipitate after UV irradiation may be stalled at DNA lesions. However, within the population of RNAPII molecules that we immunoprecipitated, there is an increased fraction that is stalled at DNA lesions. To directly demonstrate this, we have added slot blot data to the revised manuscript showing that the DNA that is co-immunoprecipitated with RNAPII after UV irradiation is enriched for CPDs (Figure 1b; and below), suggesting that we do capture a pool of RNAPII that is stalled at lesions. In line with this, we can only detect the association of CSB, CSA, UVSSA, and TFIIH subunits with RNAPII in UV-irradiated cells, but not in unirradiated cells (see Figure 1a). It is important to mention that when we perform RNA synthesis recovery assays (RRS), we detect a ~60% reduction in nascent transcripts, suggesting that a large fraction of the RNAPII pool is no longer elongating in UV-irradiated cells (see below). This will be due to a combination of stalling at DNA lesions, and repression of transcription initiation mediated by ATF3 (Epanchintsev *et al.*, *Mol Cell*, 2007).

Why the immuno-fluorescence technique established by Volker et al could not locate any Pol II /CSB while you IP a large amount of Pol II and CSB?

We would like to point out that CSB actually is recruited to sites of local UV damage using the technique established by Volker *et al.*, *Mol Cell*, 2001, but that the amount of protein that binds is much lower than what is observed for global genome repair proteins, such as XPC (Van den Boom *et al.*, *J. Cell Biol.*, 2004). We observe something very similar when we locally irradiate cells using a UV-C (266 nm) laser-irradiation system. Recruitment of XPC-GFP to UV damage is already detectable at 1% laser power, while we need much higher laser power (20%) to detect local accumulation of CSB-GFP (see figure below).

2) Why immuno-precipitations using CPD antibodies were not performed? ChIP/re ChIP investigation might demonstrate that we have stalled Pol II as a good control.

This is an excellent suggestion and we have added slot blot data to the revised manuscript, which shows that the DNA that is immunoprecipitated together with RNAPII α from UV-irradiated cells is highly enriched for CPDs (see figure 1b; and point 1).

3) You demonstrate that these factors are sequentially recruited. Control experiments using Antibodies towards these proteins would be useful to really establish the stalled complex.

We indeed use antibodies against endogenous CSB, CSA and TFIIH subunits (p89, p80, p62, cdk7) to monitor their interaction with RNAPII α in UV-irradiated cells (see for instance Figure 1a). The four antibodies we tested against UVSSA are not specific (Figure S1c), which is why we have reconstituted UVSSA knockout cells with GFP-UVSSA to monitor the association of UVSSA with RNAPII α (Figure 4).

4) In your mass spectra experiments you show that a large amount of proteins co-IP with UVSSA. Why there is no the subunits of TFIIH such as the cdk7 or Cyclin H? What about the presence of p-Tefb that phosphorylate Ser 2 of Pol II-CTD?

In our mass spectrometry approach, we found that several subunits of the RNAPII complex (3 out of 12), the TFIIH complex (2 out of 10), as well as CSB and subunits of the CSA complex could be detected to associate with UVSSA in response to UV irradiation (Figure 5B). If proteins are not detected in these mass spec approach it does not mean they do not interact. For instance, we can show in Co-IP experiments that UVSSA also associates with the p62 subunit of TFIIH (Figure 5C), which we did not detect by mass spectrometry.

To address this question, we have performed additional Co-IP experiments on RNAPII α and stained for CDK7 (TFIIH / CAK subunit), and CDK9 (p-TEFb). Our experiments show that CDK7 is recruited to damage stalled RNAPII α (Figure 1a and S1a; see below), although the signal is much weaker than what we observe for other TFIIH subunits. We could not detect CDK9 recruitment to RNAPII α after UV (see below).

5) Few years ago, it was demonstrated that TFIIH was released from Pol II once 50 nt were synthesized. In your model, TFIIH come back?

Indeed, during transcription initiation TFIIH mediates the phosphorylation of Ser5 on the carboxy-terminal domain (CTD) of the largest subunit of RNAPII through the CDK7 kinase subunit of TFIIH (Roy *et al.*, *Cell*, 1994; Shiekhattar *et al.*, *Nature*, 1995; Serizawa *et al.*, *Nature*, 1995). The core TFIIH complex is released once RNA synthesis is initiated and DSIF is recruited (Compe *et al.*, *Nat. Commun.*, 2019). Following initiation, the CDK9 subunit of pTEFb phosphorylates Ser2 on the CTD of the largest subunit of RNAPII, which triggers the release of RNAPII into active elongation (Vos *et al.*, *Nature*, 2018; Marshall *et al.*, *J. Biol. Chem.*, 1996).

In our experiments, we used an antibody that recognizes the elongating form of RNAPII that is phosphorylated at Ser2. This pSer2 modification of RNAPII is established after the release of the TFIIH complex during transcription initiation, which explains why we do not detect an interaction between the pSer2-modified form of RNAPII and TFIIH subunits in unirradiated cells (Figure 1a). Following UV irradiation, we detect that TFIIH associates with RNAPII in a manner that fully depends on CSB, CSA, and UVSSA (Figure 5d) to initiate transcription-coupled nucleotide excision repair (Figure 7c). Therefore, TFIIH indeed comes back to RNAPII to function as a repair factors rather than a transcription factor. This is now discussed in our revised manuscript.

Could you comment the CAK presence?

Previous work has shown that the CAK module is released from core TFIIH and that this switches TFIIH from a transcription into a repair factor in nucleotide excision repair (Coin *et al.*, *Mol Cell.*, 2008). Recent structural studies fully support this model by revealing that the CAK module inactivates the XPD helicase activity required for NER (Kokic *et al.*, *Nat Commun.*, 2019) explaining why dissociation of the CAK module is required for repair.

Our Co-IP experiments show that the CDK7 subunit of the CAK module is recruited to RNAPII after UV (see also point 4). We speculate in the discussion of the revised manuscript that TFIIH is recruited during repair in an inactive form, and that release of the CAK module is also required during TCR to activate TFIIH and initiate repair.

In the Ebright and Coulombe models, TFIIH is located in front of Pol II. How its binding might occur in the case of stalled Pol II? Look at Cramer (Pol II in front of DNA lesion) and Nogales paper transcription complex).

Earlier biochemical approaches (Douziech *et al.*, *Mol Cell Biol.*, 2000; Kim *et al.*, *Science*, 2000) and more recent cryo-EM studies (He *et al.*, *Nature*. 2016) have indeed shown that TFIIH is located in front of RNAPII during transcription initiation with XPB bound to the downstream DNA. In the model we propose (Figure 7c), TFIIH is also recruited to the front of RNAPII by CSB, CSA, and UVSSA to initiate repair. This model is also compatible with available structural data showing RNAPII stalled at a CPD does not undergo a conformational change (Brueckner *et al.*, *Science*. 2007), and that RAD26, the yeast orthologue of CSB, is bound to the upstream DNA behind RNAPII (Xu *et al.*, *Nature*. 2017). We hope that the reviewer appreciates that our model in Figure 7c is based on these published RNAPII structures. The study from the Nogales lab is now cited in the discussion.

6) What about the CUL4 and Roc protein that might ubiquitinate Pol II?

We have performed additional Co-IP experiments and find that DDB1, CUL4, and ROC/RBX1 all associate with RNAPII α in a UV-dependent manner (see Figure 1a and S1a; and below). We show in Figure 1g that the association of DDB1 with RNAPII α after UV is fully depend on CSA.

7) In your paper, you did not comment or mention papers from Proietti (EMBO, 2006) and Epantchisev (Mol Cell, 2017). How these findings fit (if any) in your model?

Earlier and more recent studies have revealed that CSB has an important role in transcription restart at gene promoters after UV irradiation by mediating the degradation of the UV-induced transcription repressor ATF3 resulting in the restart of RNA synthesis (Proietti-De-Santis *et al.*, EMBO J., 2006; Epantchintsev *et al.*, Mol Cell. 2017). We think that CSB has a dual role in DNA repair and transcription restart. In DNA repair, our findings suggest that CSB associates with elongating RNAPII molecules that stall at DNA lesions, which is followed by the sequential recruitment of CSA, UVSSA, and TFIIH to DNA damage-stalled RNAPII. In our model, we propose that RNAPII is subsequently removed leaving TFIIH bound to DNA to mediate downstream steps in repair, including mediating the recruitment of XPA and XPG. Following repair, CSB is also involved in transcription restart from the promoter (Proietti-De-Santis *et al.*, EMBO J., 2006; Epantchintsev *et al.*, Mol Cell. 2017). We now briefly comment on this in the introduction of our revised manuscript.

Minor points

• Could you precise the role of illudin?

We have now provided background information on illudin S. This is a natural compound from Jack 'o' Lantern mushrooms (*Omphalotus illudens*) causing DNA lesions that are exclusively repair by TCR (Jaspers *et al.*, DNA Repair. 2002).

• Fig 1f why no GFP-CSB in the input?

We have increased the brightness for this input sample, which now shows GFP-CSB signal in the input (see Figure 1f; and below). Please note that we IP on the solubilized chromatin fraction, which explains why we do not detect GFP-CSB in the chromatin input of unirradiated cells, while we can clearly detect chromatin-bound CSB in UV-irradiated cells.

• Some figs should be reduced: Fig 1a;

We have removed this figure panel in our revised manuscript.

Reviewer #2 (Remarks to the Author):

Transcription-coupled repair (TCR) is a sub-pathway of the nucleotide excision repair (NER) that removes bulky lesions, for example generated by UV light, in the transcribed regions of the genome. Although proteins that function in NER are known, their assembly mechanism at sites of DNA lesions remains poorly understood.

In this manuscript, van der Weegen et al. report an analysis of the TCR protein assembly to stalled RNA pol 2 in a series of isogenic U2OS cell lines. Experiments in this manuscript are well controlled and carefully conducted, and the findings provide a relevant step forward in mechanistic understanding of TC-NER. The authors focused on the core TC-NER proteins CSA, CSB, UVSSA and TF2H. They performed a series of IPs in different TC-NER isogenic cell lines to follow the assembly of these proteins to stalled RNA pol 2 using western blotting. The main drawback of this study is the limited number of proteins the authors could probe for by WB. The study would have benefited greatly if the assembly of different complexes would have been followed by quantitative MS. This would make the study more comprehensive and interesting to the wider spectrum of readers. A second major issue is that XR-seq was performed in KO cells and not in cells reconstituted with mutants that the authors identified in the study. It is not surprising that CSA and UVSSA KO impair NER. If authors expand on these aspects by performing quantitative interactomics and XR-seq to make the study more comprehensive, I support the publication in Nature Communications.

Main comments:

1a) The approach used in Figure 1 to study protein complexes assembled at UV-stalled RNA pol2 should be combined with quantitative MS to gain a more comprehensive view of recruited proteins.

The reviewer suggests using quantitative MS after immunoprecipitation of endogenous RNAPII-Ser2 to identify recruited proteins. This is an interesting suggestion, but not trivial, since native IP's are not always compatible with MS.

To address this point, we have attempted the following:

1. *Native IP (using crosslinked antibody) followed by on-bead digestion*

- This procedure resulted in the identification of only IgG peptides due to the digestion of the antibody used for IP (see chromatography below). We confirmed by western blot analysis that antibody crosslinking was efficient, but even the crosslinked antibody is efficiently digested on the beads. The identification success was only 4% and we did not identify CSB, which is the most abundant UV-induced interactor of RNAPII. Therefore, this procedure, which we successfully use for MS after IP of GFP-tagged proteins is not suitable after IP of endogenous RNAPII.

2. *Native IP (using crosslinked antibody), followed by running and excising proteins from a gel, followed by in-gel digestion.*
- We excised gel fragments avoiding the region where the heavy and light chain of the IgG run to avoid contamination
 - MS runs after isolation of the peptides have very low yield and we do not identify a lot of peptides.
 - From 3 biological replicates (with 2 technical replicates within each experiment), we only identified peptides from 10 proteins that were enriched after UV in Ser2-RPB1 precipitates, including CSB / ERCC6 and DDB1 (see Figure below)
 - Similarly, we only identified 2 proteins after IP of Ser5-RPB1, including CSB / ERCC6 (See Figure below)

Therefore, combining our new approach to monitor TCR complex assembly with quantitative MS is not very straightforward. We will keep optimizing this approach, but this is beyond the scope of the current manuscript. We prefer not to include these MS datasets because of their limited added value due to the small number of identified proteins.

1b) The authors state that the IP was done with an antibody against phosphor-Ser2 to follow only proteins assembled at elongating and not initiating POL2. However, there is no data presented to show that these proteins are indeed recruited specifically to the elongating POL2 after UV. The authors should use MS to compare the proteins recruited to phosphor-ser5 and ser2 after UV irradiation of cells.

See point 1a. Our IP-western approach is currently more sensitive than quantitative MS for some of the selected proteins. Therefore, to address this point without relying on MS, we performed endogenous Co-IP experiments using pSer2-RPB1 and pSer5-RPB1 antibodies. Using our approach, we detect a UV-specific association of CSA, CSB and TFIIH with both pSer2-RPB1 and pSer5-RPB1 by western blot analysis. We would also like to point out that pSer5-RPB1 is not specific for initiating RNA pol II, but is also found on elongating RNA pol II that is marked by pSer2. In line with this notion, immunoprecipitation with pSer2-RPB1 antibodies also strongly enriched for pSer5-RPB1, and vice versa. These new IP experiments are shown in Figure S1b; see below).

2) The authors should test how the interaction profile of the different mutants changes after UV using quantitative MS. In this regard, the authors could pull down GFP-tagged CSB deltaCIM, CSA deltaCIR and UVSSA deltaTIR from respective isogenic lines and compare them to GFP-tagged WT proteins. We have performed MS analyses on all the requested TCR proteins and mutants (CSB^{WT}, CSB^{ΔCIM}, UVSSA^{WT}, UVSSA^{ΔCIR}, UVSSA^{ΔTIR}). Volcano plots for all these datasets are shown below and in Figures S6 and S8. The MS data largely confirms the Co-IP experiments analysed by western blot.

To enable the reader of our paper to browse these MS datasets interactively, we developed an online application called VolcaNoseR, which is freely available (<https://huygens.science.uva.nl/VolcaNoseR/>). Our revised manuscript contains a table (see below) with hyperlinks to these interactive Volcano plots. We also added these hyperlinks to the relevant figure legends.

Figure	Link
S6a	CSB WT vs GFP-NLS
S6b	CSBΔCIM vs GFP-NLS
S6c	CSB WT vs CSBΔCIM
5b	GFP-UVSSA vs GFP-UVSSA +UV
S8a	UVSSA WT vs GFP-NLS
S8b	UVSSAΔCIR vs GFP-NLS
S8c	UVSSAΔTIR vs GFP-NLS
S8d	UVSSA WT vs UVSSAΔCIR
S8e	UVSSA WT vs UVSSAΔTIR

3) The same mutants should be also tested in XR-seq in order to define the importance of identified interaction surfaces in TC-NER

We performed XR-seq on UVSSA knockout cells complemented with either UVSSA^{WT}, UVSSA^{ΔCIR} or UVSSA^{ΔTIR}. Complementation with the wild-type proteins restores the preferential repair of CPDs in the transcribed strand of active genes, while stable expression of either the CIR or TIR mutant of UVSSA does not (Figure S9; see below). These findings confirm the importance of the identified interaction surfaces in UVSSA in transcription-coupled repair.

4) Can the authors show with recombinant purified proteins in in vitro assays the importance of identified CSA CIR and UVSSA TIR region for interaction between CSA, UVSSA and TF2H? Have the authors checked with purified proteins that CSB does not bind to UVSSA?

We appreciate this suggestion, but we would like to point out that monitoring the interaction between all these purified proteins, and their mutants, in a fully reconstituted system is technically not possible at the moment.

The reviewer asks if it is possible to confirm the interactions between CSA – UVSSA and UVSSA – TFIIH, and to look for a possible interaction between CSB - UVSSA using purified proteins. This would be a major effort involving cloning of UVSSA wild-type and deletion mutants as well as the ten subunit TFIIH complex for insect cell expression. Together with establishing purification and in vitro interaction conditions, this would take many months of work. Generating these recombinant proteins and establishing such interactions would be a big step towards the in vitro reconstitution of TCR, which would be an independent study beyond the scope of the current work. Thus, we do not believe that addressing this in the current manuscript would be appropriate, but that our manuscripts paves the way for future interaction studies in a fully reconstituted system.

Minor comments:

- Scheme in Figure 1A is confusing and not understandable without reading the manuscript text. Why is the antibody depicted before IP? This should be made easier to follow for the reader.

We removed the scheme from the paper. The antibody against RNAPII α is added together with benzonase to solubilize chromatin. Subsequently, we add beads to bind the antibody and perform the actual pull-down.

- Why are there no error bars in some lines of the clonogenic cell survival in Figures 1D, 3E, 4B?

We perform clonogenic survivals in technical duplicates / triplicates and perform at least two biological replicates for each condition. The variation between the clonogenic survivals shown in Figure 1D, 3E and 4B is very small, which is why the error bars are very small. All the raw data for these survivals is now provided in the main figures and as supplemental data.

- The role of TF2H in TC-NER should be better explained in the introduction
Included in the revised manuscript.

- It is not clear what the Supplementary Figure 1 shows. It seems redundant to Figure 1. Are these the replicates of the same experiment?

Figure 1 shows a composite from different individual Co-IP experiments. It is technically not possible to stain for all the proteins shown in Figure 1a on one membrane. All the individual Co-IP on which the composite is based are shown in Supplementary Figure 1a.

REVIEWERS' COMMENTS:

Reviewer #1 (Remarks to the Author):

The author answered to all my concerns. This is a very good paper.
It should be published.
May be a news and views could be performed.

Reviewer #2 (Remarks to the Author):

The revised manuscript addressed majority of my concerns and questions. It describes a number of well designed and high quality experiments to address the recruitment of proteins recruited to stalled RNA pol II and provides significant contribution to the understanding of TC-NER. I endorse the publication of the manuscript in Nature Communications.

I suggest following minor modifications to further improve the manuscript before publication:

The schematic figure of phosphorylated RNA pol II should be again added in Figure 1. The figure should be modified so that the reader can clearly follow the procedure without reading the legend.

Last sentence in the abstract: Word "reveal" should be exchanged to avoid repetition.

First sentence in the introduction: Word "our" should be omitted.

Figure 1b: The authors should specify in the figure that this is RNAPII α IP.

Figure 7c: In the model it should be made more clear on which proteins are interaction surfaces CIM, CIR and TIR present. Maybe a short legend in the figure would help.

I am surprised that IP with Ser2p of RNA pol II and MS did not yield better results. I think repeating this experiment and adding it in Figure 1 would be very beneficial for the manuscript. It could be that the amount of antibody and protein material just needs to be scaled up. I fully understand if the authors feel this is beyond the scope of this manuscript. The crosslinking of antibodies to beads should not be required if the light and heavy chain are cut out of the gel.

REVIEWERS' COMMENTS:

Reviewer #1 (Remarks to the Author):

The author answered to all my concerns. This is a very good paper.

It should be published.

May be a news and views could be performed.

We thank the reviewer for his / her support.

Reviewer #2 (Remarks to the Author):

The revised manuscript addressed majority of my concerns and questions. It describes a number of well-designed and high-quality experiments to address the recruitment of proteins recruited to stalled RNA pol II and provides significant contribution to the understanding of TC-NER. I endorse the publication of the manuscript in Nature Communications.

We thank the reviewer for his / her support.

I suggest following minor modifications to further improve the manuscript before publication:

1.) The schematic figure of phosphorylated RNA pol II should be again added in Figure 1. The figure should be modified so that the reader can clearly follow the procedure without reading the legend.

A modified version of the IP approach has been added to Figure 1a.

2.) Last sentence in the abstract: Word "reveal" should be exchanged to avoid repetition.

This has been changed.

3.) First sentence in the introduction: Word "our" should be omitted.

This has been changed.

4.) Figure 1b: The authors should specify in the figure that this is RNAPII α IP.

This is now specified.

5.) Figure 7c: In the model it should be made more clear on which proteins are interaction surfaces CIM, CIR and TIR present. Maybe a short legend in the figure would help.

This has been changed.

6.) I am surprised that IP with Ser2p of RNA pol II and MS did not yield better results. I think repeating this experiment and adding it in Figure 1 would be very beneficial for the manuscript. It could be that the amount of antibody and protein material just needs to be scaled up. I fully understand if the authors feel this is beyond the scope of this manuscript. The crosslinking of antibodies to beads should not be required if the light and heavy chain are cut out of the gel.

We thank the reviewer for this suggestion, but we indeed feel this is beyond the scope of the current manuscript.